# MEMORY AS STATE ABSTRACTION OVER HISTORY

## ABSTRACT

Reinforcement learning is provably difficult in non-Markovian environments, which motivates identifying tractable environment subclasses. Previous work has identified classes such as regular decision processes (Brafman & De Giacomo, 2024) (Subramanian et al., 2022) and approximate information states (Subramanian et al., 2022). While these works address significant properties such as tractability, they do not answer how the classes relate, or when users should prefer one class over another. We resolve this by defining finer POMDP subclasses in terms of memory and state abstractions. Considering agent memory as a temporally-extended abstraction over the agent's observation-action history, we prove that POMDP classes can be defined using traditional state abstractions, such as model-preservation, optimal value $Q^*$ preservation, and optimal policy $\pi^*$ preservation. In the process, we extend traditional state abstraction to "soft" (stochastic) abstractions and show how this kind of abstraction relates to stochastic memory. Our new POMDP classes include many traditional POMDP categories and enable us to show new relationships between existing classes and approximate variants. This new unified framework synthesizes and expands existing literature.

## 1 INTRODUCTION

Much of reinforcement learning makes the Markov assumption, which is unrealistic and restrictive for many applications. Hence, it is useful to generalize MDPs (Puterman, 1994) to partially observable POMDPs (Kaelbling et al., 1998). However, the class of POMDPs is too broad: worst-case performance is provably difficult (Papadimitriou & Tsitsiklis, 1987) (Zhang et al., 2012). Previous literature has thus studied a variety of tractable subclasses of POMDPs such as regular decision processes (Brafman & De Giacomo, 2024) (Subramanian et al., 2022) and approximate information states (Subramanian et al., 2022). But while many of these specialized POMDPs are better behaved, the subclasses themselves are typically defined without reference to one another, which means little is known about how they relate or which ones to prefer in which situations.

We show that it is possible to describe many of the existing POMDP subclasses using a single classification framework that centers around memory functions. It is widely known that any POMDP can be reframed as an MDP simply by remembering the agent's entire history. Memory functions compress the agent's history information into a fixed-size summary statistic, such as the internal state of a recurrent neural network (Wierstra et al., 2007; Hausknecht & Stone, 2015). Memory can therefore be thought of as an abstraction over agent histories that groups "similar" histories together into equivalence classes. This insight allows us to apply existing state abstraction frameworks to draw conclusions about POMDPs.

We propose a new classification scheme for POMDPs that identifies structure in the memory functions they support. In particular, for a given POMDP, we consider whether there exist memory functions that preserve or improve the agent's ability to express a Markov world model, optimal value function, and/or optimal policy. We also consider stochastic memory functions, for which we extend the state abstraction literature. We use the existence or absence of these certain types of memory functions to categorize POMDPs and show that many previously introduced POMDP subclasses can be expressed in terms of these categories. Moreover, we leverage and extend prior work on state abstractions to prove relationships among the categories.

The paper is organized as follows. Section 2 provides background on POMDPs and state abstractions. Section 3 reviews the definitions for a variety of existing POMDP subclasses. In section 4, we

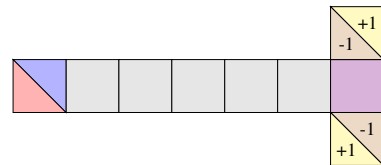
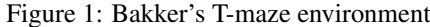

Figure 1: Bakker's T-maze environment

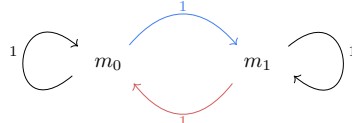

Figure 2: A $\pi^*$-optimal memory function for the T-maze.

define memory functions, introduce the notion of a POMDP's corresponding *trajectory MDP*, and show how memory functions induce state abstractions over trajectories. This connection allows us to consider memory functions that preserve the traditional state abstraction targets: $\pi^*$, $Q^*$, or model. Given any of these targets, we define three collections of memory functions: those that achieve the target optimally, "optimal"; those that improve on the target compared to having no memory whatsoever, "improving"; or those that do nothing with respect to the target, "nonimproving". Each of these targets defines classes of POMDPs based on whether such a memory function exists, and in Section 5, we prove which relationships hold between these POMDP classes. Lastly, in Section 6, we show how these classes of POMDPs systematize previously considered types of POMDPs and clarify the relationships between them.

## 2 BACKGROUND

**Markov Decision Processes (MDPs)**: In the case that an agents' environment is fully observable, we formalize an agents' environment as a tuple $\mathfrak{M} = (S, A, P, R, \gamma)$, where $S$ is the set of states, $A$ is the set of actions, $P : S \times A \to \Delta S$ is the transition function[1], $R : S \times A \to \mathbb{R}$ is the reward function, and $\gamma$ is the discount factor. An agent in the environment has a mapping $\pi : S \to \Delta A$ called the policy. We define the value function $V_{\mathfrak{M}}^\pi(s) = \mathbb{E}_{a_t \sim \pi(s_t)}[\sum_{t=0}^\infty \gamma^t r_t | s_0 = s]$ as the expected discounted return from state $s$ following policy $\pi$. We can also define the state-action value function $Q_{\mathfrak{M}}^\pi(s, a) = \mathbb{E}_{a_t \sim \pi(s_t)}[\sum_{t=0}^\infty \gamma^t r_t | s_0 = s, a_0 = a]$. MDPs have a well-defined set of optimal policies $\pi^*$, for which the optimal value functions $V^* \coloneqq V_{\mathfrak{M}}^{\pi^*}$ and $Q^* \coloneqq Q_{\mathfrak{M}}^{\pi^*}$ are constant. When the MDP $\mathfrak{M}$ in question is clear, the subscript is left off of $V^\pi$ and $Q^\pi$.

**Partially Observable MDPs (POMDPs)**: If the environment is not fully observable, we formalize the agent's decision problem as a POMDP, where the agent receives observation $\omega_t$ rather than the underlying state $s_t$. A POMDP is defined as a tuple $(S, A, P, R, \Omega, \Phi, \gamma)$, where the definitions of $S$, $A$, $P$, $R$, and $\gamma$ remain the same, and $\Omega$ is the set of observations, and $\Phi : S \to \Omega$ is the observation function. In this setting, the agent's policy depends on the sequence of observations $\omega_{1:t}$ rather than the underlying states, and at each timestep a new observation $\omega_t \sim \Phi(s_t)$ is generated.

**State Abstraction**: Given an MDP with states $S$, a state abstraction $\varphi$ is a mapping $s \mapsto x$ for $x$ in some abstract set of states $X$.[2] In this work, we consider three types of state abstractions, initially defined by (Li et al., 2006): 1) model, which preserves the one-step model, 2) $Q^*$, which preserves the state-action value function for the optimal policy, and 3) $\pi^*$, which preserves optimal actions. These state abstractions can be generalized to approximate forms and parameterized by values of $\varepsilon$ (Abel et al., 2016; Jiang, 2018). We list the definitions for exact and approximate state abstractions in Table 1.

The types of abstractions form a hierarchy, as given in Theorem 2.1. A model-preserving abstraction is necessarily a $Q^*$-preserving abstraction, and a $Q^*$-preserving abstraction is necessarily a $\pi^*$-preserving abstraction.

**Theorem 2.1** (Abstraction hierarchy from Jiang (2018)). *Let $(S, A, P, R, \gamma)$ be an MDP.*

  1. *An $(\varepsilon_P, \varepsilon_R)$-approximate model-preserving abstraction is also a $Q^*$-preserving abstraction with $\varepsilon_{Q^*} = \frac{\varepsilon_R}{1-\gamma} + \frac{\gamma \varepsilon_P R_{max}}{2(1-\gamma)^2}$.*

---

[1]$\Delta S$ denotes a probability distribution over $S$.

[2]Some authors call general maps "aggregation" and use "abstraction" to refer specifically to maps that preserve model, $Q^*$, or $\pi^*$.

| Target | Exact State Abstractions | Approximate State Abstractions |
|---|---|---|
| model | $\forall s, s'.\forall x.\forall a.\varphi(s) = \varphi(s') \Rightarrow$ $R(s,a) = R(s',a)$ $\sum\limits_{\bar{s}:\varphi(\bar{s})=x} P(\bar{s}\vert s,a) = \sum\limits_{\bar{s}:\varphi(\bar{s})=x} P(\bar{s}\vert s',a)$ | $\exists f_P : \varphi(S) \times A \to \Delta\Omega.\forall s \in S.\forall a \in A.$ $\|f_P(\varphi(s),a) - P(s,a)\|_1 < \varepsilon_P$ $\exists f_R : \varphi(S) \times A \to \mathbb{R}.\forall s \in S.\forall a \in A.$ $\|f_R(\varphi(s),a) - R(s,a)\|_\infty < \varepsilon_R$ |
| $Q^*$ | $\forall s, s'.\forall a.\varphi(s) = \varphi(s') \Rightarrow$ $Q^*(s,a) = Q^*(s',a)$ | $\exists f : \varphi(S) \times A \to \mathbb{R}.\forall s \in S.\forall a \in A.$ $\|f(\varphi(s),a) - Q^*_{\mathfrak{M}}(s,a)\|_\infty \le \varepsilon_{Q^*}$ |
| $\pi^*$ | $\forall s, s'.\exists a^*.\varphi(s) = \varphi(s') \Rightarrow$ $Q^*(s,a^*) = \max_a Q^*(s,a)$ $\max_a Q^*(s',a) = Q^*(s',a^*)$ | $\exists \pi : \varphi(S) \to \Delta A.\forall s \in S$ $\|V^{\pi\circ\varphi}_{\mathfrak{M}}(s) - V^*_{\mathfrak{M}}(s)\|_\infty \le \varepsilon_{\pi^*}$ |

Table 1: Exact and Approximate State Abstractions. $\|\cdot\|_1$ denotes the 1-norm, and $\|\cdot\|_\infty$ denotes the $\infty$-norm.

2. A $Q^*$-preserving abstraction with $\varepsilon_{Q^*}$ is also a $\pi^*$-preserving abstraction with $\varepsilon_{\pi^*} = 2\varepsilon_{Q^*}/(1-\gamma)$.

## 3 RELATED WORK

The POMDP framework was first developed for control theory in Åström (1965) and later applied to AI problems in works such as Kaelbling et al. (1998). Schmidhuber (1990), Lin & Mitchell (1992), and Meeden et al. (1993) developed the use of history features, and Lin & Mitchell (1992) and Ring (1994) developed variable-length history windows. Some approaches to memory that were previously surveyed in Kaelbling et al. (1996). McCallum (1996) imply (without explicitly mentioning "abstraction") that memory can be viewed as an abstraction over histories. However, he did not have access to the state abstraction hierarchy that came later due to Li et al. (2006).

**Related to models**: In $k$-order Markov POMDPs (Ching & Ng, 2006), the future observations are independent of the past history given the last $k$ steps of observations. If we decouple transition-model-optimality and reward-model-optimality, then we get the generalization of $k$-order Markovianity pursued by Ni et al. (2023). Those authors allowing the $k$ parameter to vary between rewards and transitions via the "reward memory length" ($k$ such that $\mathbb{E}[r_t|\tau_{1:t}, a_t] = \mathbb{E}[r_t|\tau_{t-k+1:t}, a_t]$) and "transition memory length". Efroni et al. (2022) defines the class of $k$-step decodable POMDPs, which are stronger than $k$-order Markov POMDPs. In $k$-step decodable POMDPs, the last $k$ observations can predict not only a function (the observation function) of the next state, but the state itself. If we relax the Markovianity of the environment, but still require that the environment dynamics be representable by a finite FSM, then we get the regular decision processes (RDPs) defined by Brafman & De Giacomo (2024).

**Related to policies and values**: With a finitely-transient policy of size $k$ for a POMDP defined as X, we can define a class of POMDPs that supports such a policy (Sondik, 1978). Furthermore, just as Ni et al. (2023)'s "reward" and "transition" memory lengths slightly generalized $k$-Markovianity, authors also define a "policy memory length" that generalizes finite transience in the same way. Additionally, they define a "value memory length" that corresponds in a similar way to preserving value instead of policy.

**Belief space methods**: We can also define classes of POMDPs based on the size of the reachable belief states, dependent on a specific policy, such as the optimal policy or a set of policies (Kaelbling et al., 1998). Zhang & Zhang (2001) defines "informative POMDPs", where each new observation yields information that helps partition the belief space. Roy et al. (2005) proposes a method of compressing the belief space of POMDPs with exponential family principal component analysis. Lee et al. (2007) proposes a different method that utilizes the covering number instead of PCA. Doing so, the authors achieve guarantees on the difficulty of finding approximately optimal solutions; the time required to find such a solution is polynomial in the covering number.

**Learnability**: Other related work focuses on learnability. The finite-state controller literature defines FSMs with roles similar to our memory functions, but this literature focuses on efficiently learning the controllers rather than analyzing their expressability (Poupart & Boutilier, 2003) (Junges et al., 2018) (Kara & Yüksel, 2023). Other classes of POMDPs are defined with restrictions to the

observation or transition space such that efficient learnability is guaranteed. Jin et al. (2020) defines "undercomplete" POMDPs where there are more observations than latent states. Azizzadenesheli et al. (2016) and Guo et al. (2016) both place restrictions on the allowed observation functions (full column rank), transition function (full rank), with Azizzadenesheli et al. (2016) having an additional assumption of ergodicity and Guo et al. (2016) of full reward column rank. Given these assumptions, they find efficient learning techniques.

**Problem-specific**: There are also specialized classes of POMDPs that are efficiently solvable by methods such as SLAM and Kalman filters, where constraints are placed on the transition dynamics and/or observation function, such as assuming Gaussian noise in it. Continuous POMDPs also require methods such as discretization (Kara et al., 2025).

**Generalizations**: In this paper, we focus on the standard state abstraction results because they give us a formal, quantitative hierarchy to work with. However, there are several generalizations of the abstraction concepts we use in this paper. Bisimulation metrics are generalizations of the bisimulations used to define $\mathrm{model}$-preserving abstractions (Ferns et al., 2004) (Kemertas & Jepson, 2022). They generalize approximate abstractions in a way somewhat tangential to our analysis in this paper, however, and do not admit a hierarchy. Approximate information states (AISs) model statistics that approximate the state of the environment and provide a different kind of generalization of $\mathrm{model}$-preserving abstractions (Mahajan & Mannan, 2016) (Subramanian et al., 2022) (Sinha & Mahajan, 2024) (Patil et al., 2024). Predictive representations of state (PSRs) are a different way of modeling decision problems involving compressions of possible future history predictions instead of the past history (Littman et al., 2001).

## 4 MEMORY FUNCTIONS AND STATE ABSTRACTION

A maximally expressive agent policy would have access to all previous observation-action pairs and could be defined as a map from histories, or *trajectories*, $\tau$, to a distribution over actions $\pi(\tau)$. However, it is typically intractable to have agent policies defined over the entire trajectory space $\mathcal{T}$, hence we would like to compress this space to a space of memories $M$. A simple choice for doing this is to choose a $k$-state finite state machine (FSM) (Rabin & Scott, 1959) over $M$ that transitions on each new action and observation. Formally, $\mu : M \times \Omega \times A \to \Delta M$. We call such an FSM a **memory function**, and the output $m_{t+1}$ is fed in to a policy function $\pi : M \times \Omega \to \Delta A$[3]. $m_0$ is sampled from the initial state distribution, and on each timestep, $a_t \sim \pi(m_t, \omega_t)$ and $m_{t+1} \sim \mu(m_t, \omega_t, a_t)$. This flow is also depicted in Figure 3.

We use FSM-based memory in this work as it produces a good model of systems like RNNs; RNNs with quantization are precisely FSMs with many states and complicated dynamics. Furthermore, the action of a memory function on the POMDP can be cleanly described as augmenting the state and observation space (Allen et al., 2024). We allow the memory function $\mu$ to be stochastic as stochastic memory functions are provably more powerful for goals such as improving expected return (see Subsection 5.2).

### 4.1 ABSTRACTION FUNCTION DEFINITIONS

Given a memory function, we define properties of it in terms of properties of a state abstraction it induces. To define this state abstraction, we first introduce the base MDPs on which to define the abstractions. We have two different MDPs for this: the **trajectory MDP**, which models the best that an agent can do with access to as much information as it could obtain, and the **effective MDP** that models the best that an agent can do with only access to the current observation.

**Definition 4.1** (Trajectory MDP). Given a POMDP $(S, A, P, \gamma, \Omega, \Phi, R)$ with initial state distribution $s_0$, we define the trajectory MDP to be $\mathfrak{M}' = (\mathcal{T}, A, P', R', \gamma)$, where $\mathcal{T} := \{\tau \in (\Omega \times A)^* \times \Omega\}$[4] is the space of observation-action partial trajectories, $P' : \tau \times a_t \mapsto \tau \oplus a_t \oplus \omega_{t+1}$ with $\omega_{t+1} \sim \mathbb{P}(\cdot|\tau, a_t)$ and $\oplus$ denoting concatenation, and $R'(\tau = (\omega_0, a_0, \ldots, \omega_t), a_t) := \mathbb{E}_{s_t|\tau}[R(s_t, a_t)]$. This decision process is Markov by definition as a trajectory $\tau_t$ up to time $t$ being

---

[3]Because the output of the memory function is considered to be its current memory state, it is formally a Moore machine (Moore et al., 1956)

[4]$*$ denotes the Kleene star operation, i.e. having 0 or more of these tuples in the trajectory sequence.

a prefix of $\tau_{t+j}$ implies that $\mathbb{P}(\tau_{t+k}|\tau_t, \tau_{t-1}, \dots) = \mathbb{P}(\tau_{t+k}|\tau_t)$. We also define the next observation distribution as $P'_\omega : \mathcal{T} \times A \to \Delta\Omega$ with $P'_\omega(\tau, a) \coloneqq \mathbb{P}(\omega_{t+1}|\tau, a)$. The observation-only construction was previously considered by Timmer & Riedmiller (2009) and Hong et al. (2023).

**Definition 4.2** (Effective MDP). Given a POMDP $(S, A, P, \gamma, \Omega, \Phi, R)$, we define the effective MDP to be $\widehat{\mathfrak{M}} = (\Omega, A, \widehat{P}, \widehat{R}, \gamma)$, where $\widehat{P}(\omega'|\omega, a) \coloneqq \sum_{s,s' \in S} \Phi(\omega'|s')P(s'|s,a)\,\mathbb{P}(s|\omega)$ models marginalized observation-observation transitions, and $\widehat{R}(\omega, a) \coloneqq \sum_{s \in S} R(s, a)\,\mathbb{P}(s|\omega)$, where $\mathbb{P}(s|\omega)$ is policy-dependent and describes how each hidden state $s \in S$ contributes to the overall environment behavior when we see observation $\omega$.

Now given a memory function $\mu : M \times A \times \Omega \to \Delta M$, we can define state abstraction functions $\varphi$ on top of either of these MDPs. For the trajectory MDP case, an abstraction $\varphi$ is well-defined when there exists a map $\varphi$ such that the diagram in Figure 3 commutes. We can define such a $\varphi$ as:

$$\varphi\big((\omega_0)\big) \coloneqq (m_0, \omega_0); \quad \varphi\big((\dots, \omega_t, a_t, \omega_{t+1})\big) \coloneqq \big(\mu(\mu(\dots \mu(m_0, a_0, \omega_1), \dots), a_t, \omega_{t+1}), \omega_{t+1}\big)$$

i.e., defining the abstraction to follow single steps forward in the memory function. For the effective MDP, we define the state abstraction function to have trivial memory, i.e. $\varphi(\omega) \coloneqq (m_0, \omega)$ where the memory $m$ is constant.

## 4.2 TYPES OF STATE ABSTRACTIONS

Now that we know how memory functions induce abstractions, we want to quantify how good memory functions are using these abstractions. Figure 4 shows the general relationships between abstractions and memory functions with respect to some target metric, but there are two useful kinds of memory to draw attention to: **improving** memory functions that improve over having no memory at all, and **optimal** memory functions improve. A memory function is $\varepsilon$-optimal if the abstraction it induces has targets $\varepsilon$-close to optimal. A memory function is $\varepsilon$-improving if it is *not* $\varepsilon$-nonimproving, where a memory function is $\varepsilon$-nonimproving if the abstraction it induces has targets $\varepsilon$-close to the so-called effective MDP, which models a memoryless agent (Allen et al., 2024). Just as the trajectory MDP plays a role in measuring optimality, the effective MDP plays a parallel role in measuring a lack of improvement.

In Table 2, we adapt the approximate state abstraction definitions (Table 1) for POMDPs with memory, i.e. abstracting either the trajectory MDP $\mathfrak{M}'$ or the effective MDP $\widehat{\mathfrak{M}}$. This change requires a few mathematical details we elaborate on in Appendix B. Most significantly, we allow $\varphi$ to be stochastic (i.e. for an MDP, it would map $S \to \Delta X$), and this requires us to extend function composition in the natural way to work with distributions. The definitions of $Q^*$ and $\pi^*$ abstractions are the same here as in the standard state abstraction case (Table 1). For model we make an adjustment so that a good abstraction represents Markov predictions with respect to the underlying MDP. [5] We show in Appendix E that this definitional change preserves the state abstraction hierarchy.

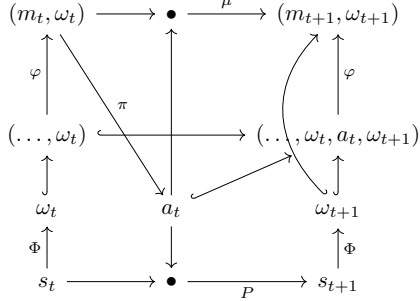
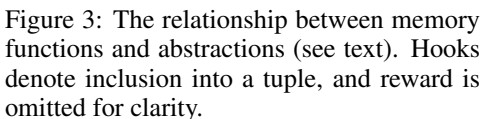
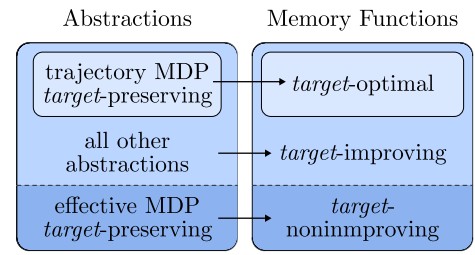

Figure 3: The relationship between memory functions and abstractions (see text). Hooks denote inclusion into a tuple, and reward is omitted for clarity.

Figure 4: The relationships between types of memory functions and types of trajectory abstractions. We elaborate on the types of abstractions in Appendix F.

From defined classes of memory functions, we define classes of POMDPs, given four attributes: *stochasticity* (stochastic or deterministic), *quality* (optimal, improving, nonimproving), *target*

---

[5]Jiang (2018) assumed $\pi : \varphi(S) \to A$ to be deterministic, but the results extend to stochastic $\pi$ as well.

| Type | Optimal | Improving |
|------|---------|-----------|
| model | $\exists f_P : \varphi(\mathcal{T}) \times A \to \Delta\Omega. \forall \tau \in \mathcal{T}. \forall a \in A$ $\|f_P(\varphi(\tau), a) - P'_\omega(\cdot\|\tau, a)\|_1 \leq \varepsilon_P$ $\exists f_R : \varphi(\mathcal{T}) \times A \to \mathbb{R}. \forall \tau \in \mathcal{T}. \forall a \in A$ $|f_R(\varphi(\tau), a) - R'(\tau, a)| \leq \varepsilon_R$ | $\forall f_P : \varphi(\mathcal{T}) \times A \to \Delta\Omega. \exists \tau \in \mathcal{T} \exists a \in A$ $\left\|f_P(\varphi(\tau), a) - \widehat{P}(\cdot\|\omega, a)\right\|_1 > \varepsilon_P$ $\forall f_R : \varphi(\mathcal{T}) \times A \to \mathbb{R}. \exists \tau \in \mathcal{T} \exists a \in A$ $\left|f_R(\varphi(\tau), a) - \widehat{R}(\omega, a)\right| > \varepsilon_R$ |
| $Q^*$ | $\exists f : \varphi(\mathcal{T}) \times A \to \mathbb{R}. \forall \tau \in \mathcal{T}. \forall a \in A$ $|f(\varphi(\tau), a) - Q^*_{\mathfrak{M}'}(\tau, a)| \leq \varepsilon_{Q^*}$ | $\forall f : \varphi(\mathcal{T}) \times A \to \mathbb{R}. \exists \tau \in \mathcal{T}. \forall a \in A$ $\left|f(\varphi(\tau), a) - Q^*_{\widehat{\mathfrak{M}}}(\omega, a)\right| > \varepsilon_{Q^*}$ |
| $\pi^*$ | $\exists \pi : \varphi(\mathcal{T}) \to \Delta A. \forall \tau \in \mathcal{T}$ $|V^{\pi \circ \varphi}_{\mathfrak{M}}(\tau) - V^*_{\mathfrak{M}'}(\tau)| \leq \varepsilon_{\pi^*}$ | $\forall \pi : \varphi(\mathcal{T}) \to \Delta A. \exists \tau \in \mathcal{T}$ $\left|V^{\pi \circ \varphi}_{\mathfrak{M}}(\tau) - V^*_{\widehat{\mathfrak{M}}}(\omega)\right| > \varepsilon_{\pi^*}$ |

Table 2: Here, $\varphi : \mathcal{T} \to \Delta M \times \Delta\Omega$. Left: The definitions of $\varepsilon$-optimal memory function, using $\mathfrak{M}'$ from Definition 4.1. Right: The definitions of $\varepsilon$-improving memory functions, using $\widehat{\mathfrak{M}}$ from Definition 4.2. $\omega$ is the last observation in $\tau$.

(model-preserving, $Q^*$-preserving, or $\pi^*$-preserving), and *number* of states $(1, k)$. **Note:** Here, "improving" means 0-improving ($\varepsilon = 0$), and "nonimproving" means 0-nonimproving. This is the setting in which we present our later results.

**Definition 4.3.** A POMDP is called **stochastic** $k$-memory *target*-optimal if it admits a corresponding $k$-state memory function that is also **stochastic** and *target*-optimal.

We likewise have definitions for *target*-improvable and *target*-nonimprovable memory functions, as well as **deterministic** memory functions.

# 5 RELATIONSHIPS BETWEEN CLASSES OF MEMORY FUNCTIONS

Next, we consider the relationships between the classes of POMDPs. We want to know, in particular, when a POMDP admitting a memory function in one class implies it admits a memory function in another class.[6]

For all other parameters held constant, we consider the following three particular families of results:

1. State abstraction results: we consider the relationships between model, $Q^*$, and $\pi^*$ in Subsection 5.1.

2. We consider the relationships between types of stochasticity (stochastic, deterministic) and size of memory function in Subsection 5.2.

3. We consider if optimal implies improving or vice versa in Subsection 5.3.

We conjecture that these are the only families of cases that we must consider because for any other cases, the implications do not follow.

## 5.1 STATE ABSTRACTION

The first family of cases, the relationships between model, $Q^*$, and $\pi^*$ given that *stochasticity*, *number*, and optimal/improving are held constant, are covered by the abstraction hierarchy. Namely, an $\varepsilon$-model-optimal memory function implies the existence of a $\frac{\varepsilon_R}{1-\gamma} + \frac{\gamma \varepsilon_P R_{\max}}{2(1-\gamma)^2}$-$Q^*$-optimal memory function, and an $\varepsilon$-$Q^*$-optimal memory function implies the existence of a $2\varepsilon_{Q^*}/(1-\gamma)$-$\pi^*$-optimal memory function. Non-trivial relationships do not appear to hold for improving memory functions.

---

[6]In our results, we ignore the "nonimproving" memory functions as every POMDP admits a trivial 1-state nonimproving blank memory function, and we consider the particular cases of 2 and an arbitrary finite number $k$ of memory states. Considering 2 memory states is important for determining if results monotically improve with increasing memory.

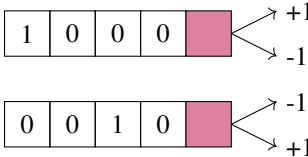

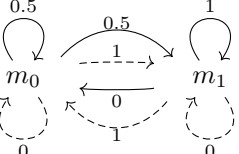

Figure 5: Two corridors, the sequence of observations of which cannot be recalled by any deterministic 2-state memory function.

Figure 6: $m_0$ is the initial state, the solid line gives transitions upon observations of $0$, and the dashed line gives transitions upon observations of $1$.

The abstraction hierarchy and these two bounds were presented earlier in the background Section 2. However, our work requires slight extensions because we consider soft approximate abstractions. The $Q^*$-preservation implies $\pi^*$-preservation proof follows directly (see Appendix E for details), and for model-preservation implies $Q^*$-preservation we require a modified lemma because we use a different model error definition. See Appendix E.1 for details.

## 5.2 STOCHASTICITY AND NUMBER

Next, let's consider when admitting a deterministic/stochastic memory function of a certain size implies admitting a deterministic/stochastic memory function of another size. We prove that nontrivial relationships between these POMDP classes do not exist, with two notable exceptions:

**Theorem 5.1.** *With an unbounded memory capacity, deterministic memory can approach the expected return of finite-size stochastic memory, when rewards are bounded: Let $\mu_k^*$ be a $k$-state stochastic memory function. For any POMDP with bounded reward and all $\varepsilon$, there exists a $k'$-state memory function which achieves an expected return that is only $\varepsilon$ less than $\mu_k^*$. Furthermore, it is sufficient to choose $k' \geq k \ln(\varepsilon(1-\gamma)/R_{max})/\ln(\gamma)$ where $R_{max}$ is the bound on reward and $\gamma$ is the discount factor. See Appendix I for the proof.*

**Example 5.1.** *With a finite memory capacity, there exist POMDPs where stochastic memory is strictly more powerful than deterministic memory: Chatterjee et al. (2004) had previously shown that a stochastic policy can aid an agent in a POMDP, and Singh et al. (1994b) had demonstrated that in a certain class of games, deterministic memory can aid a stochastic policy. Sinha & Mahajan (2024) establishes that nonstationary information states can be more powerful than stationary information states. Our paper further establishes that stochastic memory can be advantageous compared to deterministic memory, irrespective of the kind of policy. We show this via counterexample. Consider the POMDP depicted in Figure 5. The agent spawns in one of two corridors and observes a sequence of binary observations, with all actions in the corridor moving to the right. At the junction (red), the agent receives a positive reward if they choose the action "up", indicating they were in the top corridor, while they receive a positive reward in the bottom corridor if they choose "down".*

*There exists no 2-state memory function capable of distinguishing the strings 1000 and 0010, and thus the agent's memory will be identical at the junction regardless of which of the 16 two-state memory functions they have. This problem is, however, resolvable with 3 states of memory (such as with the automata that count the number of 0's mod 3 since the most recent 1) or with stochastic memory, which is shown in Figure 6. Given the policy that the agent goes up given $m_0$ and down given $m_1$, the agent will receive an expected reward of $11/16 = 0.6875$, which is higher than the 0.5 expected reward given by a deterministic memory or memoryless policy.*

These two results highlight important implications in the following figures, the first of which, Figure 9, shows which implications hold for $\pi^*$. We give a reference after each implication result to a counterexample or proof in appendices G, J, or I, and relationships that hold trivially by set inclusion are marked with [SI].

Here, we also note that while Figures 7 and 8 are defined as in Table 2 with a bounded maximum over all initial trajectories,[7] Figure 9 for $\pi^*$ is defined with an expectation over initial trajectories $\tau$. We chose this as using the expectation, $\pi^*$-optimality is equivalent to expected return preservation;

---

[7]We call this the $\infty$-norm in the Appendix.

this is how we present the proofs/counterexamples. As we show in Appendix H, bounded maximum preservation implies expected case preservation, so the hierarchy as presented in Figure 10 is valid.

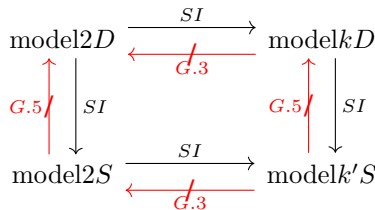

Figure 7: model

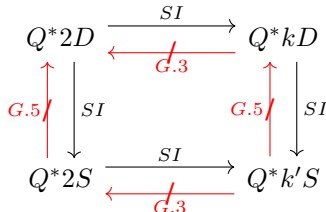

Figure 8: $Q^*$

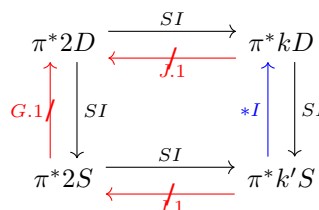

Figure 9: Expected $\pi^*$

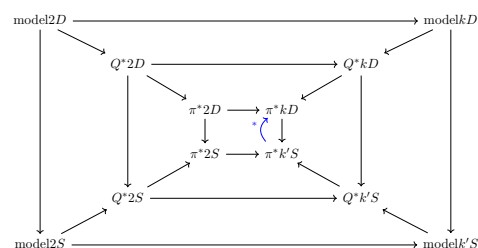

Figure 10: Hierarchy

Figure 11: The general relationships between our defined POMDP classes. We show that the same relationships hold for both optimal and improving memory functions, e.g. for both model-optimal and model-preserving. Here, 2 denotes 2-state and $k$ denotes $k$-state (with each entry potentially a different $k$), and D denotes deterministic, while S denotes stochastic. The asterisk and blue arrow mark the result from Theorem 5.1, when a converse holds.

### 5.3 IMPROVING/OPTIMAL

Lastly, we must consider when, holding *stochasticity*, *number*, and *target* constant, does *improving* imply *optimal*. Such a relationship never holds, and to show this, we can modify Figure 5 to admit a memory function that is improving but suboptimal for each target. If, at the end of the corridor, we give the agent an option of either recalling the most recent observation or recalling the entire sequence of observations, with a higher reward given to the harder task, then the agent requires only 2 states of memory to complete the easier task and improve on a memoryless agent, while the agent requires at least 3 states to have optimal memory. This example also works for $Q^*$ and model following similar reasoning.

## 6   DISCUSSION

Now that we have established new classes of POMDPs using state abstraction, we can re-organize the classes discussed in Section 3. In Tables 3 and 4, each entry represents the class of POMDPs that admits the object described, or has the property described. For example, "finite transition-model optimal FSA" means the class of POMDPs that admit a transition-model-optimal finite state machine. As the right hand side arrows indicate, latter rows are contained within earlier rows; each row is a subset of the row above it. Roughly, the top entry comprises "anything that can be considered an abstraction," the middle entry is the finite version of this, and the bottom entry is the finite Markov version of this.

In Table 3, the right column is the intersection of the left and middle columns. For example, the regular decision processes are the intersection of the two classes our paper defines, namely those with finite-state transition-model and reward-model optimal memory functions. In Table 4, we copy over the right column of Table 3 and define a corresponding set of three entries for $Q^*$ and $\pi^*$. The state abstraction hierarchy says we have horizontal implications as indicated by arrows. For example, a regular decision process necessarily admits a finitely transient policy.

| Transition-model | + | Reward-model | $\implies$ | Model |
|---|---|---|---|---|
| transition-preserving abstraction | | reward-preserving abstraction | | model-preserving abstraction |
| finite transition-model optimal FSM (e.g. memory traces with finite precision (Eberhard et al., 2025)) | | finite reward-model optimal FSM | | regular decision processes (Brafman & De Giacomo, 2024) |
| finite transition memory length (Ni et al., 2023) | | finite reward memory length (Ni et al., 2023) | | k-order Markov |

Table 3: Comparison of different models and abstractions

| Model | $\implies$ | Value | $\implies$ | Policy |
|---|---|---|---|---|
| model-preserving abstraction | | $Q^*$-preserving abstraction | | $\pi^*$-preserving abstraction |
| regular decision processes (Brafman & De Giacomo, 2024) | | finite $Q^*$-optimal FSM | | finitely transient $\approx$ finite $\pi^*$-optimal FSM (Sondik, 1978) |
| $k$-order Markov | | finite value memory length (Ni et al., 2023) | | finite policy memory length (Ni et al., 2023) |

Table 4: Comparison of Model, Value, and Policy-based Abstractions

**Other classes mentioned in Section 3**: First, in Table 3, finite approximations of memory traces are a special case of finite transition-model optimal FSMs Eberhard et al. (2025). If one maintains a memory trace in its full generality, it is not a compression of the history because it requires maintaining an unbounded decimal expansion. Thus in practice people would need to approximate memory traces with a finite decimal expansion, but this would be equivalent to a (large) $k$-state memory function. Second, by a theorem showing that approximate model-preserving abstractions are a special case of approximate information states, we see that these are broader than the "model-preserving abstraction" category and FSM-based memory functions (Subramanian et al., 2022).

**Our paper's new contributions:**

1. A new view on memory: while some previous literature considers memory as approximate belief states over an MDP, we instead suggest it can be viewed as a formal state abstraction over the trajectory MDP. (See Section 4)

2. Prove general relationships between classes of POMDPs defined with state abstraction, and generalize these classes to improving and approximate/soft variants, which have only been directly considered in the literature for certain cases. (Section 5)

3. Show how classes of POMDPs considered in related works fit into this state-abstraction-centric view. Several relationships are novel corollaries of the state abstraction hierarchy, such as the ones concerning regular decision processes. (Section 6)

## 7 CONCLUSION

We prove that memory functions induce state abstractions over trajectory MDPs and leverage this fact to define a POMDP classification framework based on traditional state abstraction targets: $\pi^*$-, $Q^*$-, and model-preservation. We define new POMDP subclasses based on whether, for any of these targets, a deterministic or stochastic memory function of a certain size exists that is optimal or improves the target. We describe how existing POMDP subclasses relate to our framework, prove which inclusions hold between them, and generalize them with $\varepsilon$-approximate variants.

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

# A   TABLE OF SYMBOLS IN THE MAIN PAPER

| Symbol | Meaning |
| --- | --- |
| $S$ | State space |
| $A$ | Action space |
| $P : S \times A \to \Delta S$ | Transition matrix |
| $R : S \times A \to \mathbb{R}$ | Reward function |
| $\gamma$ | Discount factor |
| $\Omega$ | Observation space |
| $\Phi : S \times A \to \Omega$ | Observation function |
| $M$ | Memory space |
| $\pi : M \times \Omega \to A$ | Policy |
| $\mu : M \times A \times \Omega \to \Delta M$ | Memory function |
| $k$ | Number of states in $\mu$ |
| $m$ | A memory state |
| $X$ | Abstract state space |
| $\varphi : S \to X$ | State abstraction function |
| $Q^*$ | Optimal $Q$ value function |
| $\pi^*$ | Optimal policy |
| $\varepsilon$ | Error parameter |
| $\varepsilon_P$ | Transition error parameter for model-preserving abstraction |
| $\varepsilon_R$ | Reward error parameter for model-preserving abstraction |
| $\varepsilon_{Q^*}$ | Optimal value error parameter for model-preserving abstraction |
| $\varepsilon_{\pi^*}$ | Optimal policy error parameter for model-preserving abstraction |
| $\varphi(S)$ | A subset of $X$. Allows for $f$ in lifted definitions to only be defined on the required subset of $X$, rather than the whole subset of $X$. |
| $\|\cdot\|_1$ | The 1-norm, $\|v\|_\infty = \sum_i |v_i|$. |
| $\|\cdot\|_\infty$ | The $\infty$-norm, $\|v\|_\infty = \max_i |v_i|$. |
| $\mathcal{T}$ | Space of trajectories |
| $\tau$ | A particular trajectory |
| $\mathfrak{M}'$ | Trajectory MDP |
| $P' : \mathcal{T} \times A \to \mathcal{T}$ | The transition function of the trajectory MDP |
| $R' : \mathcal{T} \times A \to \mathbb{R}$ | The reward function of the trajectory MDP |
| $P'_\omega : \mathcal{T} \times A \to \Omega$ | The distribution over next observations |
| $\widehat{\mathfrak{M}}$ | Effective MDP |
| $\widehat{P} : \Omega \times A \to \Delta\Omega$ | Transition function of effective MDP |
| $\widehat{R} : \Omega \times A \to \mathbb{R}$ | Reward function of effective MDP |
| $f, f_P, f_R, \pi$ | |
| $R_{\max}$ | The maximum possible reward (when applicable) |
| $\Delta$ | the space of distributions over a set |
| $\sim$ | sampling from a distribution |
| $\oplus$ | concatenation |
| $*$ | Kleene star in the context of the trajectory MDP definition |

Table 5: Table of symbols for the main paper (not appendix)

## B  STATE ABSTRACTION DETAILS

| Type | Optimal | Improving |
|---|---|---|
| model | $\exists f_P : \varphi(\mathcal{T}) \times A \to \Delta\Omega. \forall \tau \in \mathcal{T}. \forall a \in A$ $\|f_P(\varphi(\tau), a) - P'_\omega(\cdot\|\tau, a)\|_1 \le \varepsilon_P$ $\exists f_R : \varphi(\mathcal{T}) \times A \to \mathbb{R}. \forall \tau \in \mathcal{T}. \forall a \in A$ $\|f_R(\varphi(\tau), a) - R'(\tau, a)\| \le \varepsilon_R$ | $\forall f_P : \varphi(\mathcal{T}) \times A \to \Delta\Omega. \exists \tau \in \mathcal{T} \exists a \in A$ $\left\| f_P(\varphi(\tau), a) - \widehat{P}(\cdot\|\omega, a) \right\|_1 > \varepsilon_P$ $\forall f_R : \varphi(\mathcal{T}) \times A \to \mathbb{R}. \exists \tau \in \mathcal{T} \exists a \in A$ $\left| f_R(\varphi(\tau), a) - \widehat{R}(\omega, a) \right| > \varepsilon_R$ |
| $Q^*$ | $\exists f : \varphi(\mathcal{T}) \times A \to \mathbb{R}. \forall \tau \in \mathcal{T}. \forall a \in A$ $\|f(\varphi(\tau), a) - Q^*_{\mathfrak{M}'}(\tau, a)\| \le \varepsilon_{Q^*}$ | $\forall f : \varphi(\mathcal{T}) \times A \to \mathbb{R}. \exists \tau \in \mathcal{T}. \forall a \in A$ $\left| f(\varphi(\tau), a) - Q^*_{\widehat{\mathfrak{M}}}(\omega, a) \right| > \varepsilon_{Q^*}$ |
| $\pi^*$ | $\exists \pi : \varphi(\mathcal{T}) \to \Delta A. \forall \tau \in \mathcal{T}$ $\left| V^{\pi \circ \varphi}_{\mathfrak{M}}(\tau) - V^*_{\mathfrak{M}'}(\tau) \right| \le \varepsilon_{\pi^*}$ | $\forall \pi : \varphi(\mathcal{T}) \to \Delta A. \exists \tau \in \mathcal{T}$ $\left| V^{\pi \circ \varphi}_{\mathfrak{M}}(\tau) - V^*_{\widehat{\mathfrak{M}}}(\omega) \right| > \varepsilon_{\pi^*}$ |

Table 6: Copy of Table 2. Here, $\varphi : \mathcal{T} \to \Delta M \times \Delta\Omega$. Left: The definitions of $\varepsilon$-optimal memory function, using $\mathfrak{M}'$ from Definition 4.1. Right: The definitions of $\varepsilon$-improving memory functions, using $\widehat{\mathfrak{M}}$ from Definition 4.2. $\omega$ is the last observation in $\tau$.

Here we elaborate on the mathematical details of these abstractions.

*Epsilon*: For $\varepsilon$-optimal memory functions we want to measure the closeness to the trajectory MDP and so the error is upper bounded by $\varepsilon$. For $\varepsilon$-improving memory we instead lower bound the error which ensures strictly better performance over the effective MDP.

*Abstraction details*: We require the abstractions to preserve the present observation. In other words, for $\varphi : (\Omega \times A)^* \times \Omega \to \Delta M \times \Omega$, $\varphi$ can be written as the product of a function $(\Omega \times A)^* \times \Omega \to \Delta M$ and the function $\omega \mapsto \delta_\omega$ mapping the present observation to the point distribution at $\omega$. This resolves definitional issues that arise with considering more general memory functions and matches the intuition that the agent should always be able to perceive the present observation.

*Soft abstractions*: We allow for stochastic memory functions because they are strictly more powerful (see Example 5.1, also Deterministic memory is a special case of stochastic memory where all probabilities are 0 or 1). Because of this, we must allow also for "soft" abstractions $\varphi : S \to \Delta X$. Concretely, a stochastic memory function defines an abstraction map. If the abstraction map were purely deterministic, then given some trajectory $\tau$, the $(m_t, \omega_t)$ pair it corresponds to would be fixed, which means the memory function would actually be deterministic; hence the abstraction maps must be stochastic. Soft abstractions were previously considered by Singh et al. (1994a) and Sorg & Singh (2009). To accommodate soft abstractions, we implicitly use an extended form of function composition; for a function $g$ defined on $X$, we define $(g \circ \varphi)(s) := \mathbb{E}_{x \sim \varphi(s)} f(\cdot\|x) = \sum_x \varphi(x\|s) f(\cdot\|x)$ (Likewise, for $g : X \times A \to \mathbb{R}$, $(g \circ \varphi)(s, a) := \mathbb{E}_{x \sim \varphi(s)} f(\cdot\|x, a)$.)

## C  LIMITATIONS

As discussed in Subsection 5.2, we consider only the expected case of $\pi^*$-optimality and the $\infty$-norm case of $Q^*$-optimality and model-optimality. We anticipate other asymptotic or numerical results may hold about approximations, but we do not show those here. Additionally, our results are for the $\varepsilon = 0$ case of strict optimality or improvement, while potentially more sophisticated results could follow from considering variable $\varepsilon$.

## D  BROADER IMPACTS

The theory presented in this paper could potentially be applied to any areas of sequential decision making or reinforcement learning, such as robotics. However, it is foundational research not tied to particular applications or deployments, and thus has no societal impacts beyond what any work towards reinforcement learning theory entails. Impact control is the same as for any of these works.

# E    ABSTRACTION HIERARCHY

In order to accommodate soft abstractions, we make the change in function composition described in Appendix B. With this change, the definitions of approximate $\pi^*$-preservation and $Q^*$-preservation remain the same as in Jiang (2018), where we take the space of trajectories as the state space.

However, the definition of model-preservation requires a slight modification from Jiang's definition. Jiang defines a model-preserving abstraction to satisfy an approximate bisimulation condition: for all $s, s' \in S$ with $\varphi(s) = \varphi(s')$ and for all $a \in A$,

$$|R(s, a) - R(s', a)| \leq \varepsilon_R$$

$$\|\Phi P(s, a) - \Phi P(s', a)\|_1 \leq \varepsilon_P$$

However, this definition does not necessarily generalize well to the case that $\varphi(s)$ is a distribution. We could instead require closeness between $\varphi(s)$ and $\varphi(s')$, but it seems easier to instead take a slightly different definition, namely a result that Jiang proves as a corollary of the approximate bisimulation condition.

Jiang proves in his Lemma 3 that when the approximate bisimulation condition holds, there exists an MDP $M_\varphi = (\varphi(S), a, P_\varphi, R_\varphi, \gamma)$ such that for all $s \in S$ and $a \in A$,

$$|R_\varphi(\varphi(s), a) - R(s, a)| \leq \varepsilon_R$$

$$\|P_\varphi(x, a) - \Phi P(s, a)\|_1 \leq \varepsilon_P$$

The actual definitions of $R_\varphi$ and $P_\varphi$ do not matter for latter proofs in the hierarchy (for model-preservation to imply $Q^*$-preservation); only their existence. Importantly, this definition works naturally for soft abstractions, which will allow us to work with stochastic memory functions. This is therefore the basis of how we define model-preservation in our paper.

Here's how we derive our definitions: The conclusion of Lemma 3 is precisely that if the bisimulation condition is met, then there exists a $P_\varphi : X \times A \to \Delta X$, $R_\varphi : X \times A \to \mathbb{R}$ such that for all $s \in S$ and $a \in A$,

1. $\|P_\varphi(x, a) - \Phi P(s, a)\|_1 \leq \varepsilon_P$
2. $|R_\varphi(\varphi(s), a) - R(s, a)| \leq \varepsilon_R$

Renaming these components, we get

1. $\exists f_P : X \times A \to \Delta X. \forall s \in S. \forall a \in A \|f_P(\varphi(s), a) - (\Phi \circ P)(s, a)\|_1 < \varepsilon_P$
2. $\exists f_R : X \times A \to \mathbb{R}. \forall x \in X. \forall a \in A. |f_R(\varphi(s), a) - R'(s, a)| < \varepsilon_R$

and additional substitutions of $\varphi(S)$ for $X$ and $P'_\omega$ for $\Phi \circ P$ yields

1. $\exists f_P : \varphi(S) \times A \to \Delta\varphi(S). \forall s \in S. \forall a \in A. \|f_P(\varphi(s), a) - P'_\omega(s, a)\|_1 < \varepsilon_P$
2. $\exists f_R : \varphi(S) \times A \to \mathbb{R}. \forall s \in S. \forall a \in A. |f_R(\varphi(s), a) - R'(s, a)| < \varepsilon_R$

This is very close to our definition of model-preservation, which we recall below:

1. $\exists f_P : \varphi(\mathcal{T}) \times A \to \Delta\Omega. \forall \tau \in \mathcal{T}. \forall a \in A. \|f_P(\varphi(\tau), a) - P'_\omega(\tau, a)\|_1 < \varepsilon_P$
2. $\exists f_R : \varphi(\mathcal{T}) \times A \to \mathbb{R}. \forall \tau \in \mathcal{T}. \forall a \in A. |f_R(\varphi(\tau), a) - R'(\tau, a)| < \varepsilon_R$

The differences are that we need to substitute $\mathcal{T}$, the space of trajectories, in for $S$, and we need to replace $\Omega$ with $\varphi(\mathcal{T})$. Additionally, the output of $f_P$ would be $\varphi(\mathcal{T}) = \Delta M \times \Omega$, not $\Delta\Omega$, with this substitution. We change the output space of $f_P$ because there is no need to predict next memory states; predicting only next observations should be considered the definition.

In the following subsections, we prove that the abstraction hierarchy continues to hold.

## E.1   model-PRESERVATION IMPLIES $Q^*$-PRESERVATION

Consider an MDP with states $X = M \times \Omega$, actions $A$, transitions $(m', \omega') = x' \sim P_M(x, a) = (\mu(x, a), f_P(x, a))$ defined by the standard memory update but with averaging over observation transitions following $f_P$, and rewards given by $f_R$. Note that the transitions are stochastic and their probabilities can be written as $P_M(x'|x, a)$.

Because this is an MDP, we can define a Q-value function $f^* : \varphi(\mathcal{T}) \times A \to \mathbb{R}$ for it that satisfies the optimal Bellman equation:

$$f^*(x, a) = f_R(x, a) + \gamma \max_{a' \in A} \mathbb{E}_{x' \sim P_M(x, a)} f^*(x', a')$$

Given that this holds for all $x$, we can take an expectation of both sides over $x \sim \varphi(\tau)$ to get

$$\mathbb{E}_{x \sim \varphi(\tau)} f^*(x, a) = \mathbb{E}_{x \sim \varphi(\tau)} f_R(x, a) + \gamma \max_{a' \in A} \mathbb{E}_{x \sim \varphi(\tau)} \mathbb{E}_{x' \sim P_M(x, a)} f^*(x', a')$$

We now rewrite the expectation over $f^*(x', a')$ in terms of $\tau'$ for the given $\tau$ and $a'$. For this we will define a new function $P_T : T \times A \to \Delta T$, such that $P_T(\tau, a) = \tau \oplus (a, \mathbb{E}_{x \sim \varphi(\tau)} f_P(x, a))$.

$$\mathbb{E}_{x \sim \varphi(\tau)} \mathbb{E}_{x' \sim P_M(x, a)} f^*(x', a')$$

$$= \mathbb{E}_{x \sim \varphi(\tau)} \sum_{x'} P_M(x'|x, a) f^*(x', a')$$

$$= \mathbb{E}_{x \sim \varphi(\tau)} \sum_{\omega'} \sum_{m'} P_M((m', \omega')|x, a) f^*((m', \omega'), a')$$

$$= \mathbb{E}_{x \sim \varphi(\tau)} \sum_{\omega'} \sum_{m'} f_P(\omega'|x, a) \mu(m'|x, a) f^*((m', \omega'), a')$$

$$= \mathbb{E}_{x \sim \varphi(\tau)} \sum_{\omega'} f_P(\omega'|x, a) \sum_{m'} \mu(m'|x, a) f^*((m', \omega'), a')$$

Note that for a given $\tau$ and $a'$ the sum over $\omega'$ is equivalent to a sum over $\tau'$ with $\omega'$ being the last observation. Specifically, we can rewrite $\sum_{\omega'} f_P(\omega'|x, a)$ as an expectation over $\tau'$ as follows:

$$= \mathbb{E}_{x \sim \varphi(\tau)} \sum_{\tau'} f_P(\tau'_{\text{last } \omega}|x, a) \sum_{m'} \mu(m'|x, a) f^*((m', \tau'_{\text{last } \omega}), a')$$

Now we notice that for a given $\tau'$ and $x$, $(m', \tau'_{\text{last } \omega})$ is simply $\varphi(\tau')$ and $\mu(m'|x, a)$ is equivalent to $\mathbb{P}[m'|\varphi(\tau')]$ by the definition of $\varphi$.

$$= \mathbb{E}_{x \sim \varphi(\tau)} \sum_{\tau'} f_P(\tau'_{\text{last } \omega}|x, a) \sum_{m'} \varphi(m'|\tau') f^*((m', \tau'_{\text{last } \omega}), a')$$

$$= \sum_{\tau'} \mathbb{E}_{x \sim \varphi(\tau)} f_P(\tau'_{\text{last } \omega}|x, a) \sum_{m'} \varphi(m'|\tau') f^*((m', \tau'_{\text{last } \omega}), a')$$

$$= \mathbb{E}_{\tau' \sim P_T(\tau, a)} \sum_{m'} \varphi(m'|\tau') f^*((m', \tau'_{\text{last } \omega}), a')$$

$$= \mathbb{E}_{\tau' \sim P_T(\tau, a)} \mathbb{E}_{(m', \omega') \sim \varphi(\tau')} f^*((m', \omega'), a')$$

$$= \mathbb{E}_{\tau' \sim P_T(\tau, a)} \mathbb{E}_{x' \sim \varphi(\tau')} f^*(x', a')$$

Let $f^*(\tau, a) = \mathbb{E}_{x \sim \varphi(\tau)} f^*(x, a)$. Assuming that rewards are bounded, i.e. there exists an $R_{\max}$ such that $|R| < R_{\max}$, we can compute $\|f^*(\tau, a) - Q^*(\tau, a)\|$ as:

$$\|f^*(\tau, a) - Q^*(\tau, a)\|$$

$$= \left\| E_{x \sim \varphi(\tau)} f_R(x, a) - R(\tau, a) + \gamma \max_{a' \in A} \mathbb{E}_{\tau' \sim P_T(\tau, a)} [f^*(\tau', a)] - \max_{a' \in A} \mathbb{E}_{\tau' \sim P(\tau, a)} [Q^*(\tau', a')] \right\|$$

$$\leq \left\| \mathbb{E}_{x \sim \varphi(\tau)} f_R(x, a) - R(\tau, a) \right\| + \gamma \left\| \max_{a' \in A} \mathbb{E}_{\tau' \sim P_T(\tau, a)} [f^*(\tau', a)] - \max_{a' \in A} \mathbb{E}_{\tau' \sim P(\tau, a)} [Q^*(\tau', a')] \right\|$$

$$\leq \varepsilon_R + \gamma \max_{a' \in A} \left\| \mathbb{E}_{\tau' \sim P_T(\tau, a)} [f^*(\tau', a)] - \mathbb{E}_{\tau' \sim P(\tau, a)} [Q^*(\tau', a')] \right\|$$

$$\leq \varepsilon_R + \gamma \max_{a' \in A} \left\| \sum_{\tau' \in \mathcal{T}} [P_T(\tau'|\tau, a) f^*(\tau', a)] - \sum_{\tau' \in \mathcal{T}} [P(\tau'|\tau, a) Q^*(\tau', a')] \right\|$$

$$= \varepsilon_R + \gamma \max_{a' \in A} \left\| \sum_{\tau' \in \mathcal{T}} P_T(\tau'|\tau, a) f^*(\tau', a) - P(\tau'|\tau, a) Q^*(\tau', a') \right\|$$

$$= \varepsilon_R + \gamma \max_{a' \in A} \left\| \sum_{\tau' \in \mathcal{T}} P_T(\tau'|\tau, a) f^*(\tau', a) - P(\tau'|\tau, a) f^*(\tau', a) \right.$$

$$\left. + P(\tau'|\tau, a) f^*(\tau', a) - P(\tau'|\tau, a) Q^*(\tau', a') \right\|$$

$$= \varepsilon_R + \gamma \max_{a' \in A} \left\| \sum_{\tau' \in \mathcal{T}} (P_T(\tau'|\tau, a) - P(\tau'|\tau, a)) f^*(\tau', a) + P(\tau'|\tau, a) (f^*(\tau', a) - Q^*(\tau', a')) \right\|$$

$$< \varepsilon_R + \gamma \max_{a' \in A} \left\| \sum_{\tau' \in \mathcal{T}} (P_T(\tau'|\tau, a) - P(\tau'|\tau, a)) f^*(\tau', a) \right\|$$

$$+ \max_{a' \in A} \left\| \sum_{\tau' \in \mathcal{T}} P(\tau'|\tau, a) (f^*(\tau', a) - Q^*(\tau', a')) \right\|$$

which follows from the triangle inequality. The next inequality follows from $\|\max \cdot\| \leq \max \|\cdot\|$.

$$\leq \varepsilon_R + \gamma \max_{a' \in A} \left\| \sum_{\tau' \in \mathcal{T}} (P_T(\tau'|\tau, a) - P(\tau'|\tau, a)) f^*(\tau', a) \right\|$$

$$+ \max_{a' \in A, \tau' \in \mathcal{T}} \|f^*(\tau', a) - Q^*(\tau', a')\|$$

$$\leq \varepsilon_R + \gamma \max_{a' \in A} \left( \left( \sum_{\tau' \in \mathcal{T}} \|P_T(\tau'|\tau, a) - P(\tau'|\tau, a)\| \right) \max_{\tau' \in \mathcal{T}} \|f^*(\tau', a)\| \right)$$

$$+ \max_{a' \in A, \tau' \in \mathcal{T}} \|f^*(\tau', a) - Q^*(\tau', a')\|$$

From the definition of model preserving we have that $\varepsilon_P > \sum_{\tau' \in \mathcal{T}} \mathbb{E}_{x \sim \varphi(\tau')} f_P(x, a) - P(\tau', a)_\omega = \sum_{\tau' \in \mathcal{T}} P_T(\tau'|\tau, a) - P(\tau'|\tau, a)$. We also have that the function $f^*$ is bounded by $R_{\max}/(1 - \gamma)$ which is obtained by considering the maximum reward $R_{\max}$ in the Bellman equation for $f^*$.

$$\leq \varepsilon_R + \gamma \max_{a' \in A} \varepsilon_P \frac{R_{\max}}{(1-\gamma)} + \max_{a' \in A, \tau' \in \mathcal{T}} \|f^*(\tau', a) - Q^*(\tau', a')\|$$

$$\leq \varepsilon_R + \frac{\gamma \varepsilon_P R_{\max}}{(1-\gamma)} + \max_{a' \in A, \tau' \in \mathcal{T}} \|f^*(\tau', a) - Q^*(\tau', a')\|$$

Letting $E(\tau, a) = \|f^*(\varphi(\tau, a) - Q^*(\tau, a)\|$ we can rewrite the inequality as,

$$E(\tau, a) \leq \varepsilon_R + \gamma \varepsilon_P R_{\max} + \gamma \max_{a' \in A, \tau' \in \mathcal{T}} E(\tau', a')$$

$$\max_{a \in A, \tau \in \mathcal{T}} E(\tau, a) \leq \max_{a \in A, \tau \in \mathcal{T}} \left( \varepsilon_R + \frac{\gamma \varepsilon_P R_{\max}}{(1-\gamma)} + \gamma \max_{a' \in A, \tau' \in \mathcal{T}} E(\tau', a') \right)$$

$$\max_{a \in A, \tau \in \mathcal{T}} E(\tau, a) \leq \varepsilon_R + \frac{\gamma \varepsilon_P R_{\max}}{(1-\gamma)} + \gamma \max_{a \in A, \tau \in \mathcal{T}} \max_{a' \in A, \tau' \in \mathcal{T}} E(\tau', a')$$

$$\max_{a \in A, \tau \in \mathcal{T}} E(\tau, a) \leq \varepsilon_R + \frac{\gamma \varepsilon_P R_{\max}}{(1-\gamma)} + \gamma \max_{a \in A, \tau \in \mathcal{T}} E(\tau, a)$$

$$\max_{a \in A, \tau \in \mathcal{T}} E(\tau, a) \leq \frac{\varepsilon_R}{(1-\gamma)} + \frac{\gamma \varepsilon_P R_{\max}}{(1-\gamma)^2}$$

so for all $\tau \in \mathcal{T}, a \in A$

$$\left\| \mathbb{E}_{x \sim \varphi(\tau)} f^*(\varphi(x, a) - Q^*(\tau, a) \right\| \leq \frac{\varepsilon_R}{(1-\gamma)} + \frac{\gamma \varepsilon_P R_{\max}}{(1-\gamma)^2}$$

So the $Q^*$ error is bounded by a function of the two sources of model error.

### E.2 $Q^*$-PRESERVATION IMPLIES $\pi^*$-PRESERVATION

To prove the other implication in the hierarchy, that $Q^*$-preservation implies $\pi^*$-preservation, we can utilize the same approach is in Jiang (2018). Namely, the lemma

**Lemma E.1.**

$$\|V^* - V^{\pi_f}\|_\infty \leq \frac{2\|f - Q^*\|_\infty}{1-\gamma}$$

continues to hold in our generalized setting because the lifted function $\pi \circ \varphi$ has the same signature $S \to \Delta A$ in both of the exact abstraction and soft abstraction cases with composition defined for a soft abstraction as in Appendix B.

*Proof.* See Singh & Yee (1994). $\qquad \square$

## F ABSTRACTIONS TO MEMORY FUNCTIONS

The finest/identity abstraction maps each trajectory to its own memory state. For standard environments with unbounded trajectories, this yields an infinite-state automata memory function capable of perfect recall. The opposite of this would be an abstraction that maps all trajectories to a single memory state. This yields a trivial memory function with a single state, which is effectively always blank.

Two intermediate classes of memory functions that are useful to define are those that are $\varepsilon$-close to *target*-preserving abstractions over either the trajectory MDP or the effective MDP. The trajectory MDP models having perfect information, so being $\varepsilon$-close to *target*-preserving over it implies almost having enough information to recreate the *target*. The effective MDP models the opposite situation in which no information is available, so $\varepsilon$-close *target* preservation means having almost no advantage over a blank memory.

Anything that is not $\varepsilon$-close to *target*-preserving over the effective MDP is called "improving".

# G  COUNTEREXAMPLES

For the convenience of describing counter examples environments we refer to a *virtual environment* or a *virtual MDP*. A virtual MDP is not the true POMDP with which the agent interacts but an MDP which can be simulated by the environment. The memory agent following the optimal policy can then be described in terms of simulating the same virtual MDP in its memory and taking actions based on its internal simulation. A virtual environment is used to simplify the description of the POMDP dynamics. The reward function can be described in terms of the virtual state of the virtual MDP and the agents actions and the transition function for the environment can be expressed as taking an action in the virtual MDP in addition to other state changes needed for the POMDP. It is important to note that the agent does not necessarily *have to* represent the virtual MDP in its memory, the virtual MDP is a construction on the part of the environment which which the agent interacts.

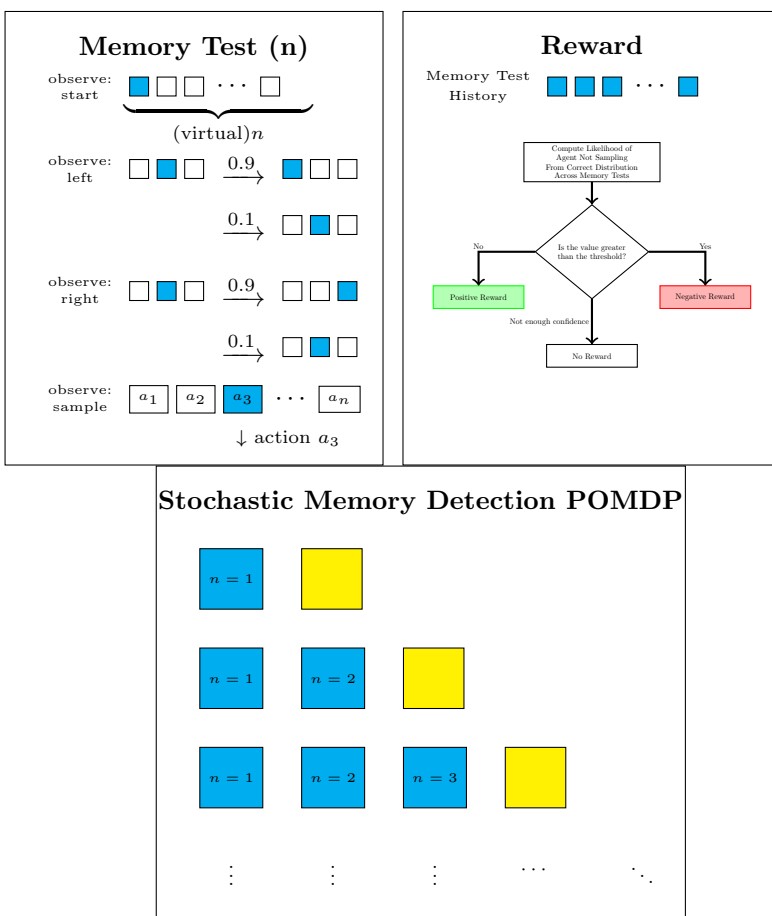

Figure 12: Example G.1 POMDP.

**Example G.1.** The following counterexample is for the following results:

- The existence of 2-stochastic expected return optimal memory doesn't imply the existence of $k$-deterministic expected return optimal memory if rewards are unbounded.

- The existence of 2-stochastic expected return improving memory doesn't imply the existence of $k$-deterministic expected return improving memory if rewards are unbounded.

- The existence of $k$-stochastic expected return optimal memory doesn't imply the existence of $k$-deterministic expected return optimal memory if rewards are unbounded.

- The existence of $k$-stochastic expected return improving memory doesn't imply the existence of $k$-deterministic expected return improving memory if rewards are unbounded.

We can prove all four results by proving a single statement: if for a given POMDP where there exists a 2-stochastic memory function which is expected return optimal then there doesn't exist a $k$-deterministic memory function which is improvable. By using the fact that improving memory functions are optimal we get results 1 and 2 and because the existence of 2 state memory implies the existence of $k$ state memory we get results 3 and 4. To prove that when there exists 2-stochastic optimal memory there doesn't exist $k$-deterministic improving memory, we now construct the following counter example and depict its structure in Figure 12.

Consider an environment where the agent's task is to simulate trajectories in a stochastic virtual MDP and to take actions which represent samples from the state distribution. The actual environment is constructed adversarially to determine if the agents actions match those expected from the distribution of virtual states in the virtual environment after some number of steps. The environment is composed of a detection mechanism and a rewarding mechanism. The detection mechanism repeats "tests", sequences of virtual actions, and analysis the agents actions to compute the likelihood of the agents memory being deterministic or stochastic. The rewarding mechanism then produces reward which is negative for deterministic memory. Finally, there is also an opt-out action that can be taken at the first time step giving the agent 0 reward. We will show that the optimal policy for any agent with deterministic memory is always to opt-out, avoiding the negative reward.

The detection mechanism is composed of a series of tests where the agent is asked to simulate a particular stochastic environment. Each test has its own virtual environment, a sequence of virtual actions, and a sequence of observations fed to the interacting agent. The virtual environment is a board with $n = 10$ spaces in a line and a token in the first space. The value of $n$ corresponds to the number of memory states of the stochastic memory function and we choose it to be 10 for the purposes of this example. For an agent with 2 stochastic memory states we would similarly have $n = 2$. Each test starts with $\omega_{\text{reset}}$ which indicates that the token should be virtually placed on the first space. Then, an arbitrary long sequence of observations from the set $\{\omega_{\text{left}}, \omega_{\text{right}}\}$ is provided to the agent. When receiving $\omega_{\text{left}}$ or $\omega_{\text{right}}$ the agent is expected to simulate the token moving left or right respectively with probability 0.9 or otherwise staying in place (the choice of probability here is arbitrary as long as it isn't uniform). Finally, the agent gets the observation $\omega_{\text{sample}}$ for which the agent is expected to take an action from $a_1, \ldots, a_{10}$ corresponding to where the simulated token ended up. For all other observations, the agent is expected to provide the action $a_0$ or else it is assumed to be deterministic by the rewarding mechanism.

After a single test the likelihood that the sampled action was in fact from the expected virtual distribution is computed and by repeating the test the statistical confidence can be increased. To run infinitely many tests infinitely many times a list of current tests is constructed and run sequentially. Upon completion a new longer test is added and the full list of tests is repeated. This ensures that in the limit as a finite time step $t$ goes to infinity, infinitely many tests, are repeated infinitely many times, and the length of the tests also approaches infinity. For example, if we currently just added the fourth test $D$ to our list of current tests $[A, B, C, D]$ the order in which the tests were evaluated from the start of the environment may look like $A, A, B, A, B, C, A, B, C, D, \ldots$.

A single test cannot be executed perfectly by an agent with deterministic memory while it is trivially handled by a stochastic memory agent with 10 memory states. The stochastic agent can sample an action from the exact distribution while the deterministic agent must in some way remember which distribution it should sample from and rely on the randomness of the policy. For any choice of finite deterministic memory size there will eventually be a test that requires remembering more possible distributions than there are memory states. In this case, the best that the agent would be able to do is sample from a distribution that is $\varepsilon$ close to the true distribution. As that particular test is repeated infinitely many times, the discrepancy between the agents sampling distribution and the true distribution will always become statistically significant and detectable. So, in the limit the probability that a deterministic agent is detected as deterministic goes to 1 and the probability that the deterministic agent is detected as stochastic goes to 0.

For the rewarding mechanism of the environment, we simply give the agent reward depending on if it is believed to have stochastic or deterministic memory. A reward that is exponential in the time step, $|R(t)| = O(1/\gamma^t)$, would be sufficient to overshadow the discount factor $\gamma$. For this

particular counter example, we will provide a positive reward for agents believed to have stochastic memory and a negative reward for agents believed to have deterministic memory (according to the detection mechanism). Positive and negative rewards are equal in magnitude at time step $t$. If the current likelihood estimate doesn't have sufficient confidence a reward of 0 is given. For this example, we wait for a confidence of at east $90\%$, although this choice is arbitrary. In the limit, the probability that the detecting mechanism is wrong about the nature of the agents memory goes to zero and so in the limit, all agents with receive positive/negative cumulative reward if they have stochastic/deterministic memory respectively.

Because an agent with a deterministic memory is always detected and gets negative reward it is optimal for all deterministic memory agents to take the opt-out action and receive 0 reward, equivalent to the best possible performance for an agent with no memory.

We now combine the detecting mechanism with the rewarding mechanism. Importantly, the detecting piece is never certain that a given agent has deterministic or stochastic stochastic memory for any finite time step $t$. This means we cannot switch to the rewarding piece indefinitely. Instead, after each test in the detecting piece we allow the rewarding piece to take over for a single time step to provide reward based on the current likelihood of the agent having deterministic memory. Because of randomness an agent with stochastic memory may be believed to have deterministic memory but in the limit this will resolve and the expected return will be positive. A deterministic memory agent with finitely many states will be detected after finitely many time steps and so will have a negative expected return even if it receives zero or positive reward for some finite number of time steps initially. Because of this, any non-stochastic agent or agent with insufficient memory will take the opt-out action when maximizing expected return and so achieve the same reward as a no memory function behavior.

Note that for this counter example we require non-finite trajectories and we only detect deterministic memory in the limit as the time step goes to infinity. If trajectories are finite, then proof 5.1 gives a deterministic memory that is better or equal to any given stochastic memory.

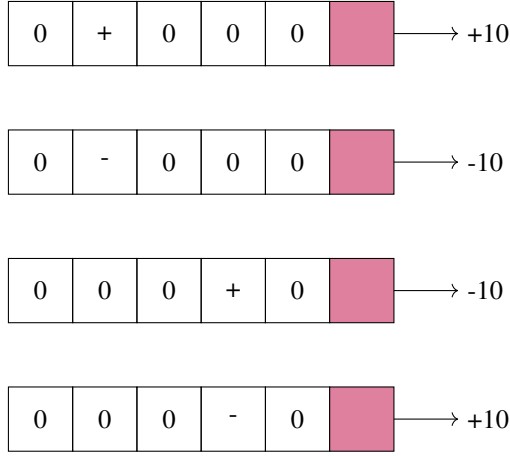

Figure 13: Four corridors.

**Example G.2.** The following counter example is for the following results:

- The existence of $k$-stochastic expected $Q^*$ optimal memory doesn't imply the existence of 2-deterministic $Q^*$ optimal memory

- The existence of 2-stochastic expected $Q^*$ optimal memory doesn't imply the existence of 2-deterministic $Q^*$ optimal memory

- The existence of $k$-stochastic expected $Q^*$ improving memory doesn't imply the existence of 2-deterministic $Q^*$ improving memory

- The existence of 2-stochastic expected $Q^*$ improving memory doesn't imply the existence of 2-deterministic $Q^*$ improving memory

This environment is depicted in Figure 13.

Consider an environment in which the agent is in one of four corridors with a single action and receives observations for 5 time steps before receiving a final reward. All observations are blank (0) with the except of one, on either the 2nd or 4th time step. That observation is either a $+$ or a $-$. If the observations seen at time step 2 are $+$ or $-$, the final reward is 10 or $-10$ respectively. If the unique observation is instead seen on the 4th time step, then the final rewards are flipped. Because the agent only has a single action at each time step, there is only one possible policy.

In this environment, an agent with $k$ memory states can track both the time step and the specific observation, $+$ or $-$, to know the exact reward that will be received at the final time step.

For an agent with 2 deterministic memory states, $m_1$ and $m_2$, there are at most $2^6$ possible memory functions. We can think of these in terms of the possible memory transitions for the three observations, $0$, $+$, and $-$, independently.

First consider the possible memory transitions for the blank observation $0$. Two options are collapsing the memory to either $m_1$ or $m_2$ in which case there is no information at the final timestep and the best possible error is 10. So for the $0$ observation we can either transpose the memory states or keep them the same (identity). For the $+$ observation we similarly cannot collapse the memory to either $m_1$ or $m_2$ or else reach the final state with a fixed memory state in corridors 2 and 4 and the best possible error is 10. The same logic follows for the $-$ memory transitions.

So we have shown that the memory function for all observations either swaps $m_1$ to $m_2$ and $m_2$ to $m_1$ or is the identity. If the transitions for the $0$ observation are the identity then we notice that for either the $-$ or $+$ observations we reach the final state with the same observation in corridors 2 and 4 or corridors 1 and 3 and so the possible error is 10. If the transitions for the $0$ observation transpose the memory state we have the same result because there is always an odd number of $0$ observations before and after any $+$ or $-$ observation.

So we have shown that for all possible 2-state deterministic memory functions, the agents memory at the final state is the same for both a corridor giving 10 reward and $-10$ reward meaning the best possible $Q^*$ error is 10, the same as no memory.

**Example G.3.** The following counter example is for the following results:

- The existence of $k$-deterministic expected $Q^*$ optimal memory doesn't imply the existence of 2-stochastic $Q^*$ optimal memory

- The existence of $k$-deterministic expected $Q^*$ optimal memory doesn't imply the existence of 2-deterministic $Q^*$ optimal memory

- The existence of $k$-stochastic expected $Q^*$ optimal memory doesn't imply the existence of 2-stochastic $Q^*$ optimal memory

- The existence of $k$-stochastic expected $Q^*$ optimal memory doesn't imply the existence of 2-deterministic $Q^*$ optimal memory

- The existence of $k$-deterministic expected $Q^*$ improving memory doesn't imply the existence of 2-stochastic $Q^*$ improving memory

- The existence of $k$-deterministic expected $Q^*$ improving memory doesn't imply the existence of 2-deterministic $Q^*$ improving memory

- The existence of $k$-stochastic expected $Q^*$ improving memory doesn't imply the existence of 2-stochastic $Q^*$ improving memory

- The existence of $k$-stochastic expected $Q^*$ improving memory doesn't imply the existence of 2-deterministic $Q^*$ improving memory

Consider an environment where the agent needs to keep track of the multiplicity of the time step. First, the agent receives a sequence of 0, 1, 2, or 3 $\omega_{\text{null}}$ observations followed by a single $\omega_{\text{end}}$ observation. At each time step, the agent can only take the action $a$. After each observation, the agent receives a reward of 0 unless the observation is $\omega_{\text{end}}$ and the time step is a multiple of 3. More specifically, here are the rewards for the following observation sequences:

1. $R(\omega_{\text{end}}) = 1$

2. $R(\omega_{\text{null}}, \omega_{\text{end}}) = 0$

3. $R(\omega_{\text{null}}, \omega_{\text{null}}, \omega_{\text{end}}) = 0$

4. $R(\omega_{\text{null}}, \omega_{\text{null}}, \omega_{\text{null}}, \omega_{\text{end}}) = 1$

Each of the possible trajectory sequences is equally likely.

A 3-state deterministic memory function is sufficient to achieve 0 reward error in this environment. Consider three states $m_1$, $m_2$, and $m_3$ that transition in a cycle with 100% probability. Whenever the memory is in state $m_1$, the initial memory state, the agent can predict a reward of 1 and otherwise predict a reward of 0. This gives a candidate $f$ which satisfies $\|[f]_\varphi - Q^*_{\mathfrak{M}}\|_\infty \le \varepsilon_{Q^*}$.

An agent with no memory could achieve a maximum error of $1/2$ by predicting a reward of $1/2$ in all cases.

We now consider the constraints that a 2-state stochastic memory function would need to satisfy in order to perform better than an agent with no memory. Note that because the agent only has one available action, we can think of $f$ as a function only of memory. We will also reduce our view to only the $\omega_{\text{end}}$ observation because if no $f$ exists which outperforms a memoryless agent over just one of the observations it also cannot exist over both. Because we are considering only a single action and a single observation, $f$ becomes a function of only the memory state. This lets us succinctly express $[f]_\varphi$ as $\mathbb{E}_{(m,\omega)\sim\varphi(\tau)}[f(m)] = \mathbb{P}(m_1|\tau)f(m_1) + \mathbb{P}(m_2|\tau)f(m_2)$. For convenience, we write $f = (f(m_1), f(m_2))$.

We now need to determine what $\mathbb{P}(m_1)$ and $\mathbb{P}(m_2)$ would be for a given trajectory. Because the observation is always $\omega_{\text{null}}$ for all time steps before the $\omega_{\text{end}}$ observation, and because the agents action is always $a$, the memory function update reduces to a function of only the previous memory state. This also lets us express it as a two by two matrix $A = \begin{pmatrix} p & 1-p \\ 1-q & q \end{pmatrix}$, where $p$ is the probability of transitioning to $m_1$ when in $m_1$ and $q$ is the probability of transitioning to $m_2$ when in $m_2$. Now, given a vector representing the probabilities of each memory state, we can find the corresponding probabilities at the next time step by multiplying this vector by $A$. Finally, we say the initial memory state distribution is $y = (y_1, y_2)$ where $y_1$ is the probability of starting in state $m_1$ and $y_2$ is the probability of starting in state $m_2$. We can now express the final memory state distribution of a trajectory of length $n$ as $yA^n$. We can then get $[f]_\varphi$, by computing $\mathbb{E}_{(m)\sim\varphi(\tau)}[f(m)] = yA^n f^T$.

Using $y$, $A$, and $f$, we can express $[f]_\varphi$, which is equivalently the final predicted reward, for the terminal states of the four possible trajectories of this environment. If we assume that this 2-stochastic memory agent is improving, we know that these predictions must be greater or less than $1/2$ based on the true $Q^*$ value.

1. $yf^T > 1/2$

2. $yAf^T < 1/2$

3. $yA^2 f^T < 1/2$

4. $yA^3 f^T > 1/2$

Note that these inequalities are strict because predicting $1/2$ would mean that the error of the agent is at least $|1/2 - 1|$ or $|1/2 - 0|$ which is not better than a no-memory agent.

From the first two conditions, we get that $yAf < yf$, and subtracting $yf = yIf$, where $I$ is the identity matrix, we get $y(A - I)f < 0$. From the second two conditions we get that $yA^2 f < yA^3 f$ and subtracting $yA^2 f$ gives $0 < y(A^3 - A^2)f$. We can calculate $A^3 - A^2$ to be $(a + b - 1)^2(A - I)$ so we get the final condition of $0 < (a + b - 1)^2 y(A - I)f$.

Because $(a + b - 1)^2$ is positive we have a contradiction. Both $0 < (a + b - 1)^2 y(A - I)f$ and $y(A - I)f < 0$ cannot be true. This implies that a 2-stochastic memory agent cannot perform any better than a no memory agent on this environment.

**Lemma G.4.** *If there exists a terminal trajectory, $\tau$, such that $|[f]_\varphi(\tau) - R(\tau)| \geq \varepsilon$ for all $f$ : $\varphi(\mathcal{T}) \times A \to \mathbb{R}$, then:*

1. $\varepsilon_{Q^*} \geq \varepsilon$
   *Because $\tau$ is a terminal trajectory $Q^*_{\mathfrak{M}}(\tau) = R(\tau)$ and by the definition of infinity norm $\|\cdot\|_\infty$, we must have that $\varepsilon_{Q^*}$ is at least $\varepsilon$*

2. $\varepsilon_R \geq \varepsilon$
   *By the definition of infinity norm $\|\cdot\|_\infty$, $\varepsilon_R$ must at least be $\varepsilon$*

**Example G.5.** The following counter example is for the following results:

- The existence of 2-stochastic Q* improving memory doesn't imply the existence of $k$-deterministic Q* improving memory.

- The existence of 2-stochastic Model improving memory doesn't imply the existence of $k$-deterministic Model improving memory.

- The existence of 2-stochastic Q* optimal memory doesn't imply the existence of $k$-deterministic Q* optimal memory.

- The existence of 2-stochastic Model optimal memory doesn't imply the existence of $k$-deterministic Model optimal memory.

First we define a virtual MDP with two states $s_1$ and $s_2$ and a parameterized set of actions $A = \{a_x | x \in [-1, 1]\}$. Actions $a_x$ with $x >= 0$ result in the the following two transitions $P(s_2|s_1, a_x) = x$, $P(s_1|s_1, a_x) = 1 - x$, and $P(s_2|s_2, a_x) = 1$. Actions $a_x$ with $x < 0$ result in the the following two transitions $P(s_1|s_2, a_x) = x$, $P(s_2|s_2, a_x) = 1 - x$, and $P(s_1|s_1, a_x) = 1$. Actions are selected uniformly at random at each time step.

We now wrap this MDP with a POMDP to produce the desired counter example. The POMDP tracks the running probability of state $s_1$ and at each time step communicates the action taken in the MDP, $a_x$, to the agent as observation $\omega_x$. At each time step the POMDP has a .1 probability of terminating and presenting the agent with the $\omega_{\text{end}}$ observation. For this observation the reward is equal to the probability of the MDP being in state $s_1$. The reward for all other observations is 0. The trajectory terminates after the $\omega_{\text{end}}$ observation. The action space for the agent is $A = \{a\}$, a single action for all time steps.

There exists a 2-stochastic optimal memory which is sufficient to predict the reward at each time step. Specifically, we take the memory function which for observations transitions its memory states $m_1$ and $m_2$ in the same way as the virtual MDP transitions its states $s_1$ and $s_2$ at each time step. This is $Q^*$ optimal. $f$ can be chosen such that $f((m_1, \omega_{\text{end}}), a) = 1$ and $f((m_2, \omega_{\text{end}}), a) = 0$ which gives an $Q^*$ error of 0 for terminal trajectories. For non-terminal partial trajectories we note that the true future return is independent of the actual time step because there is no time dependence for transitions nor termination. This lets us define $\hat{R}(s_1)$ to be the future discounted rewards if the current MDP state is $s_1$ and $\hat{R}(s_2)$ to be the future discounted rewards if the current MDP state is $s_2$. We can then choose $f((m_1, \omega_x \neq \omega_{\text{end}}), a) = \hat{R}(s_1)$ and $f((m_2, \omega_x \neq \omega_{\text{end}}), a) = \hat{R}(s_2)$ which also gives $\varepsilon_{Q^*} = 0$. This is because $P(m_1|\tau) = P(s_1|\tau)$ and $P(m_2|\tau) = P(s_2|\tau)$ so when lifting for a given trajectory $\tau$ we get $P(m_1) * f((m_1, \omega \neq \omega_{\text{end}}), a) + P(m_2) * f((m_2, \omega \neq \omega_{\text{end}}), a) = P(s_1) * \hat{R}(s_1) + P(s_2) * \hat{R}(s_2)$ which is exactly the true future discounted reward.

Following similar reasoning, this memory function is also Model optimal. For terminal trajectories $f_R$ can match $f$ and for non-terminal trajectories $f_R((\cdot, \omega \neq \omega_{\text{end}}), a) = 0$ which gives $\varepsilon_R = 0$. For

transitions, the probability of $\omega_{\text{end}}$ is always .1 and the probability of the observations $\omega_x$ follows $U(-1, 1) * .9$ which gives a natural choice of $f_P$ with $\varepsilon_P = 0$.

No memory can at best achieve $\varepsilon_{Q^*} = 1/2$ and $\varepsilon_R = 1/2$ because the true reward at the final observation can be either 0 or 1 and $f(\omega_{\text{end}}, a)$ can at best be assigned to the middle of this range to minimize error.

We now consider a $k$-deterministic memory function $\mu$, with corresponding . Lets assume that exists function $f : \varphi(\mathcal{T}) \times A \to \mathbb{R}$ such that $\|f(\varphi(\tau)) - R(\tau)\|_\infty \leq \varepsilon < 1/2$. Note that this is identical to the terminal trajectory requirement for $f$ in Lemma G.4. For simplicity we can exclude the observation and action, which are always $\omega_{\text{end}}$ and $a$ respectively, to get an identical $f : M \to \mathbb{R}$. We now prove by contradiction that such $f$ cannot exist.

For each memory state $m_i$ we define $S_i = \{\tau | \varphi(\tau) = m_i, \tau \text{ is terminal}\}$ and $R(S) = \{R(\tau) | \tau \in S\}$. Let $S$, generated by memory state $m$, be the set for which $\sup R(S) - \inf R(S) \geq \sup R(S_i) - \inf R(S_i)$ for all $S_i$. The best choice of $f(m)$ is $(\sup R(S) + \inf R(S))/2$ because for all $\tau \in S$, $|f(m) - R(\tau)| \leq (\sup R(S) - \inf R(S))/2 \leq \varepsilon < 1/2$. This implies that $\sup R(S) - \inf R(S) \leq 2\varepsilon < 1$. Either $\inf S > 0$ or $\sup S < 1$. Without loss of generality, assume that $\inf S > 0$.

We now consider an arbitrary trajectory $\tau$ and define the operation $\tau \oplus \omega_x$ for observation $\omega_x$ which generates a new terminal trajectory by inserting the observation $\omega_x$ before $\omega_{\text{end}}$ in the trajectory. Notice that for any $\tau_1, \tau_2 \in S$ and $\omega_x$, we have that $\varphi(\tau_1 \oplus \omega_x) = \varphi(\tau_2 \oplus \omega_x) = \mu(m, a, \omega_x)$. We also have that for positive $x$, $R(\tau \oplus \omega_x) = (1 - x)R(\tau)$ as defined by the probability of transitioning from $s_1$ to $s_1$ in the virtual MDP.

For any choice $0 < \varepsilon' < \frac{1}{4}\sup R$ we can choose $\tau_1, \tau_2 \in S$ and $\omega_x$ such that $\sup R(S) - R(\tau_2 \oplus \omega_x) = \varepsilon'$ and $R(\tau_1 \oplus \omega_x) < \inf R(S)$. First we pick $\tau_2 \in S$ such that $\sup R(S) - R(\tau_2) = \delta < \varepsilon'$, for $\delta \in \mathbb{R}$, which gives $R(\tau_2) = \sup R(S) - \delta$. This then means we can choose $x = 1 - (\sup R(S) - \varepsilon')/(\sup R(S) - \delta)$ which gives the desired $\sup R(S) - R(\tau_2 \oplus \omega_x) = \varepsilon'$. The condition $0 < \varepsilon' < \frac{1}{4}\sup R$ ensures $x \in (0, 1]$. We can now choose $\tau_1 \in S$ such that $R(\tau_1) - \inf R(S) = \delta' < \frac{x}{1-x}\inf R(S)$ which reduces to the desired $R(\tau_1)(1 - x) < \inf R(S)$ which is equivalent to $R(\tau_1 \oplus \omega_x) < \inf R(S)$.

We now consider a sequence of choices of $\varepsilon', \varepsilon_1, \varepsilon_2, \ldots,$ such that $\varepsilon_i = \varepsilon_{i-1}/2$. For each choice of epsilon $\varepsilon_i$ we have $\tau_{1,i}, \tau_{2,i} \in S$ and $\omega_x$ such that $\sup R(S) - R(\tau_{2,i} \oplus \omega_x) = \varepsilon'$ and $R(\tau_{1,i} \oplus \omega_x) < \inf R(S)$. Let $m_i = \varphi(\tau_{1,i} \oplus \omega_x) = \varphi(\tau_{2,i} \oplus \omega_x)$ for each $\varepsilon_i$. For the infinite sequence of $m_i$, there must be some particular $\hat{m}$ that repeats infinitely many times. Let $\hat{m}$ generate $\hat{S}$ and $I$ be the set of $\{i | m_i = \hat{m}\}$. For the pairs $\tau_{1,i} \oplus \omega_x, \tau_{2,i} \oplus \omega_x \in \hat{S}$, we have that $\inf R(\hat{S}) \leq \inf_{i \in I} R(\tau_{1,i} \oplus \omega_x) < \inf R(S)$ and $\sup_{i \in I} R(\tau_{2,i} \oplus \omega_x) = \sup R(S) \leq \sup R(\hat{S})$. This implies that $\sup R(\hat{S}) - \inf R(\hat{S}) > \sup R(S) - \inf R(S)$ which contradicts the definition of $S$. So we have that $\|f(\varphi(\tau)) - R(\tau)\|_\infty \geq 1/2$ and by Lemma G.4 we have that $\varepsilon_{Q^*} \geq 1/2$ and $\varepsilon_M \geq 1/2$.

**Example G.6.** We can consider a simpler but related counter example to G.5 to prove only the optimal cases:

- The existence of 2-stochastic Q* optimal memory doesn't imply the existence of $k$-deterministic Q* optimal memory.

- The existence of 2-stochastic Model optimal memory doesn't imply the existence of $k$-deterministic Model optimal memory.

We first define a virtual state machine with two states $s_1$ and $s_2$. The initial state is $s_1$ and at each time step there is a 10% chance of $s_1$ transitioning to state $s_2$. The state $s_2$ always transitions to itself.

We now consider a POMDP with two observations $\omega$ and $\omega_{\text{end}}$ and one action $a$ which wraps the virtual state machine. At each time step the agent receives observation $\omega$ and when the agent takes its action $a$ the virtual state machine is updated. At each time step there is a 10% chance of the trajectory terminating in which case the agent receives the $\omega_{\text{end}}$ observation and the following reward is equal to the probability of the virtual state machine being in state $s_1$. All other rewards are 0.

There exists a 2-stochastic optimal memory which is sufficient to predict the reward at each time step. Specifically, we take the memory function which for observations transitions its memory states $m_1$ and $m_2$ in the same way as the virtual MDP transitions its states $s_1$ and $s_2$ at each time step. This is $Q^*$ optimal. $f$ can be chosen such that $f((m_1, \omega_{\text{end}}), a) = 1$ and $f((m_2, \omega_{\text{end}}), a) = 0$ which gives an $Q^*$ error of 0 for terminal trajectories. For non-terminal partial trajectories the value of $f$ for each memory state can be adjusted based on the probability distribution of time steps before the environment terminates. This memory function is similarly Model optimal. For terminal trajectories $f_R$ can match $f$ and for non-terminal trajectories $f_R((\cdot, \omega \neq \omega_{\text{end}}), a) = 0$ which gives a model reward error of 0. For transitions, the probability of receiving the next observation is always fixed so the choice of $f_P$ is trivial.

Note that an agent with no memory, upon getting the terminal observation $\omega_{\text{end}}$ can at best guess the center of the range of possible rewards $[1, 0)$ and so has $\varepsilon_{Q^*} \geq .5$ and $\varepsilon_R \geq .5$ by Lemma G.4.

We now consider a $k$-deterministic memory agent. At the final observations there are an infinite number of possible rewards that the agent may receive in the range $[1, 0)$. However, because there are only $k$ memory states, $f((m_i, \omega_{\text{end}}), a)$ can only take on $k$ possible values, one for each $m_i$. By pigeon hole, there must exist at least one terminal trajectory, $\tau$ for which $|f(\tau) - R(\tau)| > 0$. This then implies that $\varepsilon_{Q^*} > 0$ and $\varepsilon_R > 0$ by Lemma G.4. So, no $k$-deterministic memory function exists.

## H  INFINITY NORM TO EXPECTED CASE

Here we explain the relationships between the infinity norm and expected cases.

First, we show model error being zero implies the agent's memory gives a Markov representation of the MDP. Suppose that the the model error definition

$$\exists f_P : \varphi(\mathcal{T}) \times A \to \Delta\Omega. \forall \tau \in \mathcal{T}. \forall a \in A. \|f_P(\varphi(\tau), a) - P'_\omega(\cdot|\tau, a)\|_1 < \varepsilon_P$$

is satisfied for $\varepsilon_P = 0$. Then,

$$\exists f_P : \varphi(\mathcal{T}) \times A \to \Delta\Omega. \forall \tau \in \mathcal{T}. \forall a \in A. f_P(\varphi(\tau), a) = P'_\omega(\cdot|\tau, a)$$

where $P'_\omega = \mathbb{P}(\cdot|\tau, a_t)$. This says that given $\varphi(\tau_t) = (m_t, \omega_t)$ and $a_t$, the agent can predict the distribution over next observations $\omega_{t+1} \sim \mathbb{P}(\omega_{t+1}|\tau_t, a_t)$ perfectly. Thus, $\mathbb{P}(\omega_{t+1}|\tau_t) = \mathbb{P}(\omega_{t+1}|m_t, \omega_t, a_t)$, which is the definition of memory yielding a Markov representation. Model error being zero implying that rewards are Markov follows similarly.

Second, we show the connection between $\pi^*$-preservation and expected return. Observe

$$\left| \mathbb{E}_\tau [V^{\pi \circ \varphi}_{\mathfrak{M}}(\tau) - V^*_{\mathfrak{M}'}(\tau)] \right| \leq \mathbb{E}_\tau \left| V^{\pi \circ \varphi}_{\mathfrak{M}}(\tau) - V^*_{\mathfrak{M}'}(\tau) \right| \leq \left\| V^{\pi \circ \varphi}_{\mathfrak{M}}(\tau) - V^*_{\mathfrak{M}'}(\tau) \right\|_\infty \leq \varepsilon_{\pi^*}$$

where the first inequality follows by Jensen's inequality. Thus, if there exists a $\pi : \varphi(\mathcal{T}) \to \Delta A$ such that the final inequality follows (which is, by definition, $\pi^*$-preservation), then for this $\pi$, the expected return is also constrained.

Third, the relationship between $Q^*$-preservation and expected value error of $\pi^*$ in the text from Sutton & Barto (2018) is similar. Recall the expected value error definition, defined in the context of function approximation, where $\mu$ is some distribution over states, $\hat{v}$ is the function approximator value function, $\omega$ is the approximated state, and $v_\pi(s)$ is the true value of state $s$ under the policy $\pi$:

$$\overline{VE}(\omega) \coloneqq \mu(s) \left[ v_\pi(s) - \hat{v}(s, \omega) \right]^2$$

and recall the $Q^*$-preservation definition:

$$\exists f : \varphi(\mathcal{T}) \times A \to \mathbb{R}. \forall \tau \in \mathcal{T}. \forall a \in A. |f(\varphi(\tau), a) - Q^*_{\mathfrak{M}'}(\tau, a)| \leq \varepsilon_{Q^*}$$

To get from $Q^*$-preservation to expected value error for $\pi^*$, we must: First, define $Q^*$-preservation with a 2-norm over outputs rather than $\infty$-norm to get

$$\exists f : \varphi(\mathcal{T}) \times A \to \mathbb{R}. \forall \tau \in \mathcal{T}. \forall a \in A. |f(\varphi(\tau), a) - Q^*_{\mathfrak{M}'}(\tau, a)|^2 \leq \varepsilon_{Q^*}$$

Second, consider only state-values instead of state-action values:

$$\exists f : \varphi(\mathcal{T}) \to \mathbb{R}. \forall \tau \in \mathcal{T}. |f(\varphi(\tau)) - V^*_{\mathfrak{M}'}(\tau)|^2 \leq \varepsilon_{V^*}$$

Third, take an expectation over $\tau$ rather than a maximum over $\tau$.

# I    EXPECTED CASE PROOFS

Here we restate Theorem 5.1 as given in the main text.

**Lemma I.1.** *Let $\mu_k^*$ be a $k$-state stochastic finite automata that will serve as a memory function in a POMDP. For any POMDP with bounded reward and for all $\varepsilon$, there exists a $k'$-state memory function which achieves an expected return that is only $\varepsilon$ less than $\mu_k^*$. Furthermore, it is sufficient to choose $k' \geq k \ln(\varepsilon(1-\gamma)/R_{max})/\ln(\gamma)$ where $R_{max}$ is the bound on reward and $\gamma$ is the discount factor.*

This result is used for the following individual results:

- The existence of 2-stochastic Expected Return improving memory doesn't imply the existence of $k$-deterministic expected return improving memory.

- The existence of 2-stochastic Expected Return optimal memory doesn't imply the existence of $k$-deterministic expected return improving memory.

*Proof.* Let $\mu_k^*$ be the given $k$-SFA memory function with the corresponding policy $\pi^*$. Let $\hat{\mu}_{k'}$ be the $k'$-state memory function with corresponding policy $\hat{\pi}$.

For a given POMDP, let $\tau_t$ be a trajectory in the environment of states, observations, memory states, actions and rewards up to time step $t$ where memory state $m_t$ is being chosen. Let the observation, memory states, and rewards for a time step $t$ be $\omega_t$, $m_t$, and $r_t$, respectively.

For a given $\tau_t$, we define $G_{\pi,\mu}(\tau_t)$ as the expected sum of discounted rewards for trajectories that start with $\tau_t$ and then proceed according to the policy $\pi$ and memory function $\mu$.

$$G_{\pi,\mu}(\tau_t) = \mathop{\mathbb{E}}_{\tau|\tau_t} \left[ \sum_{i=t}^{\infty} \gamma^{i-t} r_i \right]$$

We then define $G_{\pi,\mu}(m_t, \tau_t)$ as the expected sum of discounted rewards for trajectories that start with $\tau_t$, transition to memory state $m_t$ at time step $t$, and then proceed according to the policy $\pi$ and memory function $\mu$.

$$G_{\pi,\mu}(m_t, \tau_t) = \sum_{\tau_{t+1}} \mathbb{P}(\tau_{t+1}|\tau_t, m_t) G_{\pi,\mu}(\tau_{t+1})$$

where $\mathbb{P}(\tau_{t+1}|\tau_t, m_t)$ is the probability of a trajectory of length $t+1$ given that it starts with trajectory $\tau_t$ of length $t$ and that the memory state at time step $t$ is $m_t$ given the policy $\pi$.

Let $P(m'|m, a, \omega)$ be the probability distribution for the transitions of the memory function $\mu$. This gives $P^*(m'|m, a, \omega)$, the probability distribution of the stochastic memory function $\mu_k^*$, and $\hat{P}(m'|m, a, \omega)$, the probability distribution of the deterministic memory function $\hat{\mu}_{k'}$.

For any time step $t$ we can write the expected return of the stochastic policy as:

$$G_{\pi^*,\mu_k^*} = \mathop{\mathbb{E}}_{\tau_t} \left[ \sum_{m_t \in M} P^*(m_t|\omega_t, a_{t-1}, m_{t-1}) G_{\pi^*,\mu_k^*}(m_t, \tau_t) \right]$$

Because $M$ is finite, there must exist a $\hat{m}_t$ such that for all possible $m_t \in M$

$$G_{\pi^*,\mu_k^*}(\hat{m}_t, \tau_t) \geq G_{\pi^*,\mu_k^*}(m_t, \tau_t)$$

We then let $\hat{P}(\hat{m}_t|\omega_t, a_{t-1}, m_{t-1}) = 1$ and have $\hat{p}$ be 0 for all other $m_t$. This guarantees that

$$\mathop{\mathbb{E}}_{\tau_t} \left[ \sum_{m_t \in M} P^*(m_t|\omega_t, a_{t-1}, m_{t-1}) G_{\pi^*,\mu_k^*}(m_t, \tau_t) \right] \leq \mathop{\mathbb{E}}_{\tau_t} \left[ \sum_{m_t \in M} \hat{P}(m_t|\omega_t, a_{t-1}, m_{t-1}) G_{\pi^*,\mu_k^*}(m_t, \tau_t) \right]$$

Note that such assignment of $\hat{P}$ is equivalent to a deterministic memory function. So we have that for a particular time step $t$ the memory state can be chosen in a deterministic way to achieve the same or better expected return when compared to choosing the memory state according to $\mu_k^*$

The same argument can be made inductively, conditioning on a finite initial trajectory $\tau_{\text{start}}$. We can consider longer and longer starting trajectories and in each case we can deterministically assign $\hat{P}$ to achieve the same or better expected return when compared to $\mu_k^*$.

$$
\begin{aligned}
\mathbb{E}_{\tau_t | \tau_{\text{start}}} &\left[ \sum_{m_t \in M} P^*(m_t | \omega_t, m_{t-1}) G_{\pi^*, \mu_k^*}(m_t, \tau_t) \right] \leq \\
\mathbb{E}_{\tau_t | \tau_{\text{start}}} &\left[ \sum_{m_t \in M} \hat{P}(m_t | \omega_t, m_{t-1}) G_{\pi^*, \mu_k^*}(m_t, \tau_t) \right]
\end{aligned}
\tag{1}
$$

where $\mathbb{E}_{\tau_t | \tau_{\text{start}}}$ is the expectation over trajectories $\tau_t$ that start with $\tau_{\text{start}}$. Importantly, this holds only for finite trajectories $\tau_{\text{start}}$. Consider picking the memory states deterministically as described for trajectories $\tau_{\text{start}}$ of increasing length. Equation 1 will continue to hold and at some finite point the $G_{\pi^*, \mu_k^*}(m_t, \tau_t)$ term will become epsilon small due to the bounded reward. This means that a deterministic memory can achieve the same or better expected return compared to the stochastic memory function for a finite number of time steps $t$ and after is $\varepsilon$ close. To achieve this however, we need to distinguish identical observation, action, memory state pairs that might occur when considering trajectories of different lengths. To remedy possible conflicts that would prevent always selecting the optimal memory transitions, we can augment the memory with the current time step $t$.

We construct $\hat{\mu}_{k'}$ by making $t$ copies of each memory state in $\mu_k^*$, one for each of the first $t$ time steps. So for a given memory state $m$ from $\mu_k^*$ we now have $m_t$ for each time step t. The policy $\hat{\pi}$ can be defined to return the same action as $\pi^*$ for each of the $t$ duplicates of a given memory state, ignoring the time step. We then construct $\hat{\mu}$ as described above by taking the best choice of memory state transition at each time step ensuring that $G_{\pi^*, \mu_k^*} \leq G_{\hat{\pi}, \hat{\mu}_{k'}}$.

This gives us a $k'$-deterministic memory function with $k' = k * t$. To guarantee $G_{\pi^*, \mu_k^*} - G_{\hat{\pi}, \hat{\mu}_{k'}} < \varepsilon$ for a given $\varepsilon$ we can consider the worst case which would be a difference of $R_{\max}$ right after the first $t$ time steps. This gives the expression $R_{\max} \gamma^t (1 + \gamma + \gamma^2 + \dots) \leq \varepsilon$ which means it is sufficient to take $t$ greater than $\ln(\varepsilon(1 - \gamma)/R_{\max})/\ln(\gamma)$. This works because once the deterministic memory function matches the performance of the stochastic memory function for all trajectories of a sufficiently large finite length, all further rewards are negligibly small due to the discount factor $\gamma$.

$\square$

## J   EXPECTED CASE COUNTEREXAMPLES

**Example J.1.** The following counter example is for the following results:

- The existence of $k$-deterministic expected $\pi^*$ optimal memory doesn't imply the existence of 2-stochastic $\pi^*$ optimal memory

- The existence of $k$-deterministic expected $\pi^*$ improving memory doesn't imply the existence of 2-stochastic $\pi^*$ improving memory

Consider an environment where the agent is first shown an integer observation $\omega_i$ between 1 and $k$, and then a recall observation, $\omega_{\text{recall}}$. At the recall observation, the agent can either take the $a_{\text{exit}}$ action to receive a reward of 0, or an action $a_1, a_2, a_3, \dots a_k$ corresponding to one of the possible observations it received. If the agent selects the correct action it receives a reward of 1 and otherwise $-k$.

An agent with $k$ memory states can update the memory state based on the first observation $\omega_i$, $\mu(\cdot, \omega_i, \cdot) = m_i$. We then have the policy $\pi(m_i, \omega_{\text{recall}}) = a_i$ which achieves an expected return of 1.

Now, consider an agent with 2 stochastic memory states, $M = \{m_1, m_2\}$, which doesn't take the exit action. We have two steps that occur probabilistically, the selection of the memory state and the selection of the action. We can write the probability of a particular action in terms of the initial observation as $\mathbb{P}(a_i|\omega_j)$. When $i = j$ the agent took the correct action and gets a reward of $1$ and otherwise, it gets a reward of $-k$. This means we can write the expected return as

$$\text{Expected Return} = \frac{1}{k} \sum_{i=1}^{k} 1 * \mathbb{P}(a_i|\omega_i) - k(k-1)(1 - \mathbb{P}(a_i|\omega_i)) =$$

$$= \frac{1}{k} \sum_{i=1}^{k} \mathbb{P}(a_i|\omega_i)(1 + k(k-1)) - k(k-1)$$

$$= -k(k-1) + \frac{1 + k(k-1)}{k} \sum_{i=1}^{k} \mathbb{P}(a_i|\omega_i)$$

We have that $\mathbb{P}(a_i|\omega_i) = \mathbb{P}(a_i|m_1)\,\mathbb{P}(m_1|\omega_i) + \mathbb{P}(a_i|m_2)\,\mathbb{P}(m_2|\omega_i)$ and because $\sum_{i=1}^{k} \mathbb{P}(a_i|m) = 1$ for all $m$ we can bound $\sum_{i=1}^{k} \mathbb{P}(a_i|\omega_i) = \sum_{i=1}^{k} \mathbb{P}(a_i|m_1)\,\mathbb{P}(m_1|\omega_i) + \mathbb{P}(a_i|m_2)\,\mathbb{P}(m_2|\omega_i) \leq max_i\, \mathbb{P}(m_1|\omega_i) + \mathbb{P}(m_2|\omega_i) \leq 2$.

For $k \geq 3$ this gives:

$$\text{Expected Return} \leq -k(k-1) + \frac{1 + k(k-1)}{k} 2$$

$$\leq -k(k-1) + \frac{1}{k} + 2(k-1)$$

$$\leq \frac{1}{k} + (2-k)(k-1) \leq 0$$

This means that taking the exit action $a_{\text{exit}}$ is always optimal for the 2 stochastic memory function and this matches the expected return of no-memory.

This counter example shows that the existence of $k$-deterministic memory doesn't imply the existence of 2 stochastic. By set inclusion this also gives us that $k$-stochastic memory doesn't imply 2-stochastic memory, $k$-deterministic memory doesn't imply 2-stochastic memory, and $k$-stochastic memory doesn't imply 2-deterministic memory. Because we have that the $k$-deterministic memory is optimal and the 2-stochastic memory is not improving we have that this example extends to both the optimal and improving tables.

