# OpenReview forum: "Memory as State Abstraction Over History"
_ICLR.cc/2026/Conference — Submitted to ICLR 2026_

### Official Review · Reviewer_Y5MW · 2025-10-29

**Soundness:** 2
**Presentation:** 2
**Contribution:** 2
**Rating:** 2
**Confidence:** 3

**Summary:**

This paper aims to deepen our understanding of history abstraction in POMDPs. It analyzes abstraction classes, contrasts deterministic versus stochastic memory, and examines the role of memory capacity. The paper contributes theoretical insights into how memory functions can serve as abstractions over history, connecting perspectives from RL, control, and automata theory.

**Strengths:**

- The authors make commendable efforts to unify viewpoints from reinforcement learning, control theory, and automata theory. This interdisciplinary framing is valuable for understanding the conceptual foundations of history, abstraction, and memory.
- The derivations appear mathematically rigorous (though detailed checking would require reviewing the appendix). The formalization of memory functions and abstraction types is well structured.
- The paper is well motivated by the need to formalize the notion of memory in POMDPs, which is a critical and underexplored area.

**Weaknesses:**

**Limited novelty and missing contextualization.** The discussion of related work is insufficient to situate the paper’s originality.

For example, the concept of approximate history abstraction in Table 2 (Optimal column) directly parallels the Approximate Information State (AIS) framework, though the terminology differs. The connection becomes clear in *“Bridging State and History Representations: Understanding Self-Predictive RL”* (Ni et al., 2024), which extends exact abstraction notions ($\pi^*$-, $Q^*$-, and model-preserving) to POMDPs and unifies abstraction from a history-based perspective. Since that work, not discussed in this paper, already treats trajectory MDPs as a unifying view of POMDP abstraction, the novelty claimed here seems overstated.

In Section 6: it is ambiguous whether Tables 3 and 4 summarize prior work or present new theoretical results.
   - In *Ni et al., 2023*, Theorem 1 establishes the relationship between memory lengths and that paper also covers $k$-order MDPs.
   - In AIS and *Ni et al., 2024*, abstraction types in POMDPs are systematically derived.

Could you clarify whether your tables reproduce, extend, or reinterpret these results, and explain what the directional arrows denote?

Missing discussion of other relevant POMDP theory.  The paper overlooks works like *“When Is Partially Observable Reinforcement Learning Not Scary?”* (2022), which also classifies POMDP learnability and provides complementary theoretical framing. Including this would strengthen the contextualization and highlight the distinct contribution of this work.

**Clarity and accessibility issues.**  Figures such as Fig. 3 and Fig. 11 are difficult to interpret on first read. Simplifying the visuals or providing a more guided explanation could make the insights clearer to readers less familiar with the formalism.

**Limited significance for the ICLR threshold.**  While the theory is developed, the paper’s contribution is purely conceptual. To reach the ICLR standard, it would help to demonstrate empirical implications—e.g., experiments showing how the proposed theory informs representation design, memory capacity choice, or policy learning efficiency.

**Questions:**

- Section 5.1’s novelty seems to lie in extending the deterministic memory function from AIS to stochastic, correct?
- What exactly is “memory capacity”? Can it be formally defined?
- Could you elaborate on why there is no 2-state memory sufficient to distinguish strings `1000` and `0010` (for instance, by storing the first character)?
- Intuitively, the optimal memory can be deterministic if memory capacity is unbounded. Is the core message that stochasticity becomes essential under *finite* memory capacity constraints?

---

> ### Author Response · Authors · 2025-12-02
>
> Thank you for your review!
>
> ### Weaknesses
>
> > Limited novelty and missing contextualization
>
> There are a few significant differences between Ni et al. 2024 and our own work.
> 1. In comparing related work, Ni et al. emphasizes types of representations of the environment underlying POMDPs, while we emphasize classes of POMDPs. For example, we discuss regular decision processes and memory traces.
> 2. Our results in Section 5 focus concern the size and stochasticity of memory functions, which Ni et al. 2024 does not consider at all.
> 3. We define improving memory functions, which are not considered by Ni et al.
>
> > Clarity and accessibility issues
>
> We will fix these diagrams.
>
> ### Questions
>
> > Section 5.1’s novelty seems to lie in extending the deterministic memory function from AIS to stochastic, correct?
>
> AISs already cover the stochastic case; see definition 7 in Subramanian et al., 2021. What they do not cover is types of abstractions other than model-preserving abstractions. They contain model-preserving abstractions as a special case, but do not address the abstraction hierarchy or $Q^\*$ or $\pi^\*$ preserving abstractions.
>
> > What exactly is “memory capacity”? Can it be formally defined?
>
> "Memory capacity" here just means the number of states.
>
> > Could you elaborate on why there is no 2-state memory sufficient to distinguish strings `1000` and `0010` (for instance, by storing the first character)?
>
> A 2-state memory cannot store the first character because it can't keep track of which character was first! If it would transition to states 0 or 1 given first characters 0 or 1, respectively, then it would also transition to states 0 or 1 for the following 3 characters. The fact that these strings are indistinguishable can be shown by enumerating the 16 possible 2-state machines. This is part of the "separating words" literature (see "Remarks on Separating Words" by Demaine et al. as an example: https://arxiv.org/abs/1103.4513).
>
> > Intuitively, the optimal memory can be deterministic if memory capacity is unbounded. Is the core message that stochasticity becomes essential under finite memory capacity constraints?
>
> Yes, one conclusion you can draw is that with finite memory constraints, stochastic memory is essential in certain environments, and also helpful overall.

---

### Official Review · Reviewer_ychT · 2025-11-01

**Soundness:** 3
**Presentation:** 1
**Contribution:** 2
**Rating:** 2
**Confidence:** 2

**Summary:**

This paper proposes a unified framework of POMDP subclasses in terms of memory functions and state abstractions. First, the authors show how memory functions induce state abstractions over trajectory and effective MDPs and recontextualize traditional state abstraction relationships in this framework. Then, the authors present additional results in their framework for stochastic and deterministic memory functions. Finally, their work is shown to further unify and expand upon the existing literature.

**Strengths:**

- This paper provides a novel perspective and understanding on existing concepts in the POMDP litterature that is grounded in mathematical rigor.
- The paper gives novel insights around the stochastic-deterministic axes of memory functions, giving grounded examples and theorems around when a deterministic memory can be as powerful as stochastic ones, and how stochastic memories can be more expressive than deterministic ones
- Their framework helps unify and synthesize the existing literature in POMDPs, providing a comprehensive overview of different POMDP subclasses and how they relate to each other.

**Weaknesses:**

- The paper can be quite dense in mathematical concepts, making it difficult to follow and read, especially if the reader is not familiar with all concepts. It can be beneficial to provide more examples that ground certain concepts for a more intuitive understanding.
- While the results presented throughout the paper are interesting, it is difficult to follow the motivation and general story of each result, which seem to exist more or less in a vacuum.
- Following the previous two points, there is a lack of experimental evidence. Experiments are not only useful to empirically validate the results of a paper, but they help ground theoretical results into something that is simpler to visualize and understand for the reader, as well as provide concrete evidence as to how theoretical results can be beneficial and impact practitioners. As is, the authors provide no clear path forward for practitioners to build upon their work.
- Some of the results, for instance Table 2, appear to mostly be a re-derivation of existing results. While this can be important, especially if new results are elucidated from this re-derivation, it is not clear how these re-derivations provide additional insights beyond their original works.

**Questions:**

- The purpose of the results in Table 2 are not made clear. In line 245, the motivation seems to be to create an analogous table to Table 1, yet the columns Optimal and Improving in Table 2 are not 1-to-1 analogies to the exact state abstractions and approximate state abstractions in Table 1. The definitions here seem unnecessarily complex as well when discussing $\epsilon$-optimal $\epsilon$-improving, or $\epsilon$-nonimproving, since the results in either column are mostly the complementary side of the other column. Why is it necessary for the results in the other section to define the optimality conditions with respect to both the trajectory MDP and the effective MDP?
- The results on the stochasticity of a memory function in section 5.2 are interesting, and could benefit from empirical results, even if just in a toy environment. Does that mean that a stochastic memory function can learn to solve the environment in Figure 5 given a smaller capacity than a deterministic one?

---

> ### Author Response · Authors · 2025-12-02
>
> Thank you for your review!
>
> ### Weaknesses
>
> > The paper can be quite dense in mathematical concepts, making it difficult to follow and read, especially if the reader is not familiar with all concepts. It can be beneficial to provide more examples that ground certain concepts for a more intuitive understanding.
>
> Thank you for the suggestion. We will do this. We have added analyses of a few common POMDPs in our response to reviewer V1cN, and we will include these for illustration in the paper.
>
> > While the results presented throughout the paper are interesting, it is difficult to follow the motivation and general story of each result, which seem to exist more or less in a vacuum.
>
> We have tried to present the results in the follow order:
> 1. Introduction: discuss our results at a high level
> 2. Background: basic concepts used from the literature
> 3. Related work
> 4. Section 4: formalize new definitions that we use in our results. This defines new classes of POMDPs.
> 5. Section 5: state our new theorems that give implications between the defined POMDP classes. These are divided into three parts: re-establishing the state abstraction hierarchy, results giving implications based on stochasticity and size of the memory state machines, and results about improving and optimal memory functions.
> 6. Discussion: organize POMDP classes in the literature  using the new POMDP classes.
>
> > Following the previous two points, there is a lack of experimental evidence. Experiments are not only useful to empirically validate the results of a paper, but they help ground theoretical results into something that is simpler to visualize and understand for the reader, as well as provide concrete evidence as to how theoretical results can be beneficial and impact practitioners. As is, the authors provide no clear path forward for practitioners to build upon their work.
>
> Our paper's core result is a theoretical one which we believe is significant enough to stand on its own. Providing real experimental results would require developing a memory learning algorithm which is well outside the scope of this work. However, we do intend for our work to be clear and accessible to the reader. To help provide more clarity and motivation we include toy environments and show the memory functions they admit.
>
> Which particular sections/lines did you find difficult to follow and read?
>
> > Some of the results, for instance Table 2, appear to mostly be a re-derivation of existing results. While this can be important, especially if new results are elucidated from this re-derivation, it is not clear how these re-derivations provide additional insights beyond their original works.
>
> The results in table 2 are crucial to thinking about memory as a state abstraction. Exact state abstractions are enough for only deterministic memory. We define approximate state abstractions for stochastic memory and this allows the key idea of our work, memory as a state abstraction, to be well defined.
>
> ### Questions
>
> > The purpose of the results in Table 2 are not made clear. In line 245, the motivation seems to be to create an analogous table to Table 1, yet the columns Optimal and Improving in Table 2 are not 1-to-1 analogies to the exact state abstractions and approximate state abstractions in Table 1. The definitions here seem unnecessarily complex as well when discussing $\epsilon$-optimal $\epsilon$-improving, or $\epsilon$-nonimproving, since the results in either column are mostly the complementary side of the other column. Why is it necessary for the results in the other section to define the optimality conditions with respect to both the trajectory MDP and the effective MDP?
>
> The results *are* direct analogies with only a few changes
> 1. Replace concrete "states" with trajectories, and abstract states with $M\times\Omega$
> 2. Then the Optimal column is exactly the definition of state abstractions over the trajectory POMDP
>     3. (And we the memory functions we look at to having $\Omega$ separate from $M$)
> 4. The improving column is exactly the _opposite_ of state abstraction over the effective MDP. This is because a state abstraction over it would be *nonimproving*.
>
> > The results on the stochasticity of a memory function in section 5.2 are interesting, and could benefit from empirical results, even if just in a toy environment. Does that mean that a stochastic memory function can learn to solve the environment in Figure 5 given a smaller capacity than a deterministic one?
>
> It depends on what you mean "learn to solve". A 2-state stochastic memory function cannot achieve optimal performance in the Figure 5 environment, but it can achieve moderately better performance than without memory.

---

### Official Review · Reviewer_NyUx · 2025-11-01

**Soundness:** 3
**Presentation:** 3
**Contribution:** 3
**Rating:** 6
**Confidence:** 3

**Summary:**

This paper presents a novel view of memory in POMDPs as state abstraction in trajectory space – namely, a memory function based on a finite state machine is used to define state abstractions in memory space. The advantage of doing this is that you can then show that the typical model/Q*/pi* abstraction hierarchy holds on the memory space. The authors partition the space of memory functions into those which are optimal and improving, which describes whether the memory function represents the POMDP perfectly (in the trajectory MDP) or simply better than having no memory (relative to the effective MDP). The framework further distinguishes memory functions by their stochasticity (deterministic vs stochastic) and size (number of memory states), creating a taxonomy of POMDP subclasses. The main theorems establish relationships between these classes, and unify existing POMDP subclasses from the literature through the state abstraction lens.

**Strengths:**

- The paper presents a novel unifying framework which is conceptually elegant and allows us to view prior work through a single conceptual lens.
- The writing and mathematical exposition is mostly clear throughout, and the figures and diagrams are illustrative.
- The related work section is extensive, thorough, and helps contextualize the paper's objective.
- I believe this paper has potential to have a significant impact on both POMDP theory and the theory of abstraction, although my understanding may be a bit outdated here.

**Weaknesses:**

- Definition 4.2 lacks detail, particularly around the effective MDP construction and the role of policies (see Questions).
- The paper has no experimental results. Do the authors think it would be possible to conduct an experiment which takes some (more) realistic POMDP, e.g. a simulated environment or game, and measures what "class" it falls into? I'm not sure this is completely well-formed, but I'd be curious to understand if there's a result where we would control some salient aspect of the environment (transition structure? number of states?) and measure how the state dimensionality of the optimal memory function varies. Understanding the learnability of such a memory function could be interesting too, although it is likely beyond the scope of this work.
- As the authors point out in the limitations, only the \epsilon = 0 case is studied. It would indeed be interesting to understand the propagation of errors in the hierarchy for the \epsilon > 0 case.

**Questions:**

- On line 223, the authors state that $\mathbb{P}(s | \omega)$ is policy-dependent. This is confusing to me. What policy is being considered here, and why is it not written as part of the definition of the effective MDP? How does the choice of policy affect the downstream results?
- "We use FSM-based memory in this work as it produces a good model of systems like RNNs." What would happen if we just defined the memory function as a general mapping from full trajectories to abstract states? My impression is that the structure of the FSM is convenient for the analysis, but practical implementations might use transformers (as in Ni et al.), which have a different structure.
- Line 238: "improving memory functions that improve over having no memory at all, and optimal memory functions improve." – I think the second half of the sentence is missing.
- I was a little confused on first read about the term "target" which is used throughout. I now take this to be a placeholder for model/Q*/pi*, but this could probably be stated a bit more explicitly.
- I recognize that this paper is foundational theory work, but given the broader current interest in memory for systems such as LLMs or deep RL agents, the authors could perhaps discuss the relation between their insights and contemporary empirical problems in the field. What can we take away about the design of effective agents with memory? Where might we expect real-world POMDPs to fall within this taxonomy?

---

> ### Author Response · Authors · 2025-12-02
>
> Thank you for the thoughtful review.
>
> ### Weaknesses
>
> > Definition 4.2 lacks detail, particularly around the effective MDP construction and the role of policies (see Questions).
>
> We answer this in the Questions section below.
>
> > The paper has no experimental results. Do the authors think it would be possible to conduct an experiment which takes some (more) realistic POMDP, e.g. a simulated environment or game, and measures what "class" it falls into? I'm not sure this is completely well-formed, but I'd be curious to understand if there's a result where we would control some salient aspect of the environment (transition structure? number of states?) and measure how the state dimensionality of the optimal memory function varies. Understanding the learnability of such a memory function could be interesting too, although it is likely beyond the scope of this work.
>
> When a memory function is found that improves on the performance for one of the targets, this immediately implies an upper bound on the size of the smallest improving memory function and a lower bound on the size of the smallest optimal memory function, so we do gain information about which POMDP classes the POMDP is contained in.
>
> We are doubtful it is possible to learn which classes a POMDP is contained in more efficient than actually finding or constructing memory functions.
>
> > As the authors point out in the limitations, only the \epsilon = 0 case is studied. It would indeed be interesting to understand the propagation of errors in the hierarchy for the \epsilon > 0 case.
>
> If you mean the abstraction hierarchy, then the results for general $\epsilon$ are known! We list them in Section 5.1. The results follow from known literature for $Q^\*$ implying $\pi^\*$, while for model implying $Q^\*$, because the definition of model preservation is slightly different in our work, we must re-establish the known relationship in Appendix E.1.
>
> ### Questions
>
> > On line 223, the authors state that $P(s|\omega)$ is policy-dependent. This is confusing to me. What policy is being considered here, and why is it not written as part of the definition of the effective MDP? How does the choice of policy affect the downstream results?
>
> Thank you for catching this. Reviewer V1cN had the same question; we must amend the definition of improving memory functions to have the $>\epsilon$ inequality defining it hold for all optimal memoryles policies.
>
> > "We use FSM-based memory in this work as it produces a good model of systems like RNNs." What would happen if we just defined the memory function as a general mapping from full trajectories to abstract states? My impression is that the structure of the FSM is convenient for the analysis, but practical implementations might use transformers (as in Ni et al.), which have a different structure.
>
> The definitions of optimal and improving memory function should remain valid with more general memory functions. However, the results in Section 5 on implication between POMDP classes depend on the particular recursive structure of FSM-based memory.
>
> > Line 238: "improving memory functions that improve over having no memory at all, and optimal memory functions improve." – I think the second half of the sentence is missing.
>
> It should say that optimal memory functions are at least as good as any other memory function.
>
> > I was a little confused on first read about the term "target" which is used throughout. I now take this to be a placeholder for model/Q*/pi*, but this could probably be stated a bit more explicitly.
>
> We will add that.
>
> > I recognize that this paper is foundational theory work, but given the broader current interest in memory for systems such as LLMs or deep RL agents, the authors could perhaps discuss the relation between their insights and contemporary empirical problems in the field. What can we take away about the design of effective agents with memory? Where might we expect real-world POMDPs to fall within this taxonomy?
>
> The two main insights are that
> 1. Having memory functions be stochastic can be useful.
> 2. Memory should not necessarily be added incrementally: negative results about $k$-improving memory functions implying $2$-improving memory functions show that just because adding $k$ memory states may be useful, performance gains may not be seen by any states added less than this.

---

### Official Review · Reviewer_V1cN · 2025-11-01

**Soundness:** 4
**Presentation:** 3
**Contribution:** 3
**Rating:** 6
**Confidence:** 3

**Summary:**

The paper proposes a unified theory that treats an agent’s memory as a state abstraction over history, formalizing how finite-state (deterministic or stochastic) memory functions induce abstractions on a trajectory MDP and how a memoryless effective MDP serves as the baseline for improvement. This lets the authors define POMDP subclasses by whether a memory exists that preserves or improves classic abstraction targets, including model, Q*, and π*. The framework yields new inclusion relationships among classes (e.g., model → Q* → π* preservation under memory-induced abstractions) and sharp results on randomness vs. capacity: a finite stochastic memory can be ε-approximated by a larger deterministic memory with an explicit size bound, while for fixed capacity stochastic memory can strictly outperform deterministic (constructive example). Finally, the taxonomy systematizes and connects many prior POMDP notions, e.g., regular decision processes, -order Markov structure, finite-memory policies, and approximate information states, within one lens centered on memory-induced abstractions, clarifying when each subclass applies and how they inter-relate.

**Strengths:**

1.The paper provides a clear classification scheme for POMDPs based on memory functions, which elegantly connects many disjoint concepts in the literature. By using memory as a temporally extended state abstraction, it becomes possible to compare previously unrelated subclasses in a single framework.

2.The paper is thorough in formally defining each concept and proving relationships. All major claims are supported either by rigorous proofs or by well-chosen counterexamples.

3.The work’s perspective on treating memory as a first-class abstraction is novel in that it merges two traditionally separate lines of research: state abstraction in fully observable MDPs and memory design in POMDPs.

4.The authors systematically consider all combinations of key factors (target type, optimal vs improving, deterministic vs stochastic, memory size of 2 vs finite k vs unbounded), covering a wide range of corner cases.

**Weaknesses:**

1.the effective MDP (Definition 4.2) is defined by marginalizing states to get a model P̂(ω'|ω, a), but because P(s| ω) depends on the agent’s policy, this effective MDP is not a fixed property of the environment but rather a derived construct given a policy (or policy class). The authors intend it as the baseline of a memoryless agent’s best achievable model, but a reader might be confused about how P(s| ω) is obtained and whether the effective MDP is defined w.r.t. an optimal memoryless policy or an arbitrary one.

2.This is primarily a theoretical paper which is appropriate for the contributions, but one weakness is the absence of any empirical illustration or case study demonstrating the usefulness of the classification. For instance, it would have been insightful to take a known POMDP benchmark (maybe a simple partially observable gridworld or a Tiger problem variant) and analyze which memory class it falls into, or how an algorithm might exploit that knowledge.

3.Following the above, a brief discussion on how one might test or infer the memory needs of an environment (perhaps via training performance with increasing memory sizes, or by recognizing structural clues) would have added practical relevance.

4.Minor grammatical and spelling issues:
(1)Line 076: the case that an agents’ (agent’s) environment is fully …
(2)Line 137: Those authors allowing (allow) the k parameter…
(3)Line 160: the controllers rather than analyzing their expressability (expressibility)…
(4)Line 292: is fed in to (into) a policy function…
(5)Line 239: and optimal memory functions improve. (improve what?)
(6)Line 323: determining if results monotically (monotonically) improve…
(7)Line 1084: for a confidence of at east (least)…
(8)Line 1085: nature of the agents (agent’s) memory…
(9)Line 1286: gives an (a) Q∗ error of…
(10)Line 1086: all agents with (will) receive positive/negation…

**Questions:**

1.The framework introduces many subclasses (e.g. stochastic 2-memory model-optimal, deterministic k-memory Q*-improving, etc.). In a real application, determining which category a given problem falls into might be challenging. Do the authors have thoughts on practical tests or diagnostic methods to infer an environment’s memory class? For instance, is there a way to detect that a POMDP has a finite model-preserving memory without actually discovering the memory function?

2.Could the authors provide more intuition or a simple example of a soft abstraction? For instance, is there a scenario in a small POMDP where the optimal abstraction naturally must be stochastic (perhaps due to symmetric states that need random tie-breaking)? Understanding this would help readers appreciate why a stochastic memory function can do better than a deterministic one.

3.In Section 5.3, the authors demonstrate that “improving” does not imply “optimal” by constructing a POMDP where a small memory yields some reward but a larger memory is needed for the maximum reward. The converse question is also interesting: does having an optimal memory imply the existence of a smaller improving memory?

---

> ### Author Response · Authors · 2025-12-02
>
> Thank you for the thoughtful review.
>
> ### Weaknesses
>
> > 1.the effective MDP (Definition 4.2) is defined by marginalizing states to get a model P̂(\omega'|\omega, a), but because P(s| \omega) depends on the agent’s policy, this effective MDP is not a fixed property of the environment but rather a derived construct given a policy (or policy class). The authors intend it as the baseline of a memoryless agent’s best achievable model, but a reader might be confused about how P(s| ω) is obtained and whether the effective MDP is defined w.r.t. an optimal memoryless policy or an arbitrary one.
>
> You are right that each choice of policy defines a distribution over states $P(s|\omega)$, which defines, in turn, a set of next-state distributions, values, and policies. These are used to define the improving abstractions. To make sure our definition of improvement is independen of the policy, we require that the inequalities defining improving abstractions in Table 2 hold for the next-state distributions, values, and policies for _all_ optimal memoryless policies.
>
> > 2.This is primarily a theoretical paper which is appropriate for the contributions, but one weakness is the absence of any empirical illustration or case study demonstrating the usefulness of the classification. For instance, it would have been insightful to take a known POMDP benchmark (maybe a simple partially observable gridworld or a Tiger problem variant) and analyze which memory class it falls into, or how an algorithm might exploit that knowledge.
>
> Here are analyses of a few environments (all with deterministic memory):
> 1. Cheese maze (Chrisman, 1992)
>     - 2 memory states are necessary for optimality and improvement for all targets (model, $Q^\*$, $\pi^\*$)
> 2. T-maze (Bakker, 2001)
>     - 2 states for are necessary policy optimality
>     - 2 states for value improvement and model improvement
>     - Let $m$ be the number of corridor states. $2m$ memory states are necessary for for value and model optimality.
> 3. Tiger (Cassandra et al., 1994)
>     - infinite states are required for optimality for all targets
>     - but only 2 states are required for improvement for all targets
>
> > 3.Following the above, a brief discussion on how one might test or infer the memory needs of an environment (perhaps via training performance with increasing memory sizes, or by recognizing structural clues) would have added practical relevance.
>
> > 4.Minor grammatical and spelling issues
>
> We have fixed these in the new draft.
>
> ### Questions
>
> > 1.The framework introduces many subclasses (e.g. stochastic 2-memory model-optimal, deterministic k-memory Q*-improving, etc.). In a real application, determining which category a given problem falls into might be challenging. Do the authors have thoughts on practical tests or diagnostic methods to infer an environment’s memory class? For instance, is there a way to detect that a POMDP has a finite model-preserving memory without actually discovering the memory function?
>
> We are not sure how one would do this; our guess is that one must generally discover the memory function to know it exists.
>
> > 2.Could the authors provide more intuition or a simple example of a soft abstraction? For instance, is there a scenario in a small POMDP where the optimal abstraction naturally must be stochastic (perhaps due to symmetric states that need random tie-breaking)? Understanding this would help readers appreciate why a stochastic memory function can do better than a deterministic one.
>
> The 1000 and 0010 environment (Example 5.1) is a case where given a certain amount of finite memory, a stochastic memory function does perform better. You can design an FSM where a 1 observation mixes the state distribution, and a 0 observation pushes it towards a specific state. An agent can use this stochastic FSM to "count" how how it has been since it has observed the last 1, which allows it to achieve better-then-random performance in this environment. Because this memory function is stochastic, the abstraction induced by it is necessarily soft.
>
> > 3.In Section 5.3, the authors demonstrate that “improving” does not imply “optimal” by constructing a POMDP where a small memory yields some reward but a larger memory is needed for the maximum reward. The converse question is also interesting: does having an optimal memory imply the existence of a smaller improving memory?
>
> It does, as long as there is _any_ improving memory possible. If there is not, then the classes of optimal, improving, and nonimproving memory functions collapse.

---

### Author Response · Authors · 2025-11-12
**Missing reviews**

Dear Area Chair,

Our paper has not received any reviews as of time of writing this message. We just wanted to flag this so that it's on your radar. We are looking forward to the discussion.

---

### Meta-Review · Area_Chair_xH1o · 2026-01-07

**Summary:**

The paper introduces a classification scheme POMDPs in terms of memory functions, which allows for unifying existing POMDP subclasses in the literature and bring them into a single framework through the lens of state abstraction. This classification renders highly relevant in many application domains where non-Markovianity arises. The core idea is that memory functions could compress key information from the agent's history into a fixed-size summary statistics, which allows one to leverage memory as a tool for defining state abstractions over trajectory. This further allows for grouping similar histories and thus potentially defining relevant equivalence classes.

All reviewers agree that the paper offers a conceptually elegant and novel framework for classification of POMDPs through a relevant lens (memory functions). In particular, the paper provides novel insights. However, a key concern that still stands is lack of empirical evaluation, especially for a venue like ICLR. Some reviewers firmly believe that this limits accessibility of the results to the ICLR audience.

Besides, some concerns were related to how the method compare with some recent literature; I believe there were mostly addressed. Some reviewers raised some technical and presentational concerns, some of which were sufficiently addressed in the rebuttal. Yet the paper is not still there to convey its key message.

Overall, in view of the received reviews and reviewers' opinions, and predicting how they would change their ratings, I recommend rejection.

**Reviewer Concerns:**

__Lack of empirical evaluations.__ All reviewers agree that this is primary a theory work that offer a conceptually elegant framework. Yet, they mentioned that lack of empirical evaluations might be a key limitations here, especially for a venue like ICLR. Further, they mention that for such a mathematically dense paper, empirical evaluations could make the underlying message more accessible to the broader range of audience of ICLR.

Having read the reviews and reflecting, I believe it would be difficult to reach consensus among this matter on this matter, namely, whether for a venue such as ICLR, and for such a theory paper, empirical evaluation is key missing piece or not.  A related point is discussed next, which would help the paper convey its message.

__Lack of empirical illustrations.__ Some reviewers (e.g., V1cN) mentioned that including empirical illustration --e.g., by applying the method on some benchmark POMDPs-- could prove helpful in demonstrating the idea. The rebuttal addressed this and reported the results for 3 well-known POMDPs. The results appear promising, although limited to the case of deterministic memory.

__Question about Definition 4.2.__ Some reviewers raised questions about some key definitions (especially Definition 4.2) and concepts such as memory capacity. These were addressed well in the rebuttal.

__Which category the environment belongs.__ An important question, relevant in real applications, is which category a given problem falls into. This issue remains, as the rebuttal explains. Although this somewhat limits the practical relevance of the development,

__Missing comparison to some related literature.__ This was raised by one reviewer. The rebuttal addressed this sufficiently and clarified the positioning of its scope and contribution in relation to the mentioned literature (e.g., Ni et al. (2023; 2024)). I consider this issue as resolved. However, it's difficult to predict how the offered elaboration would impact the reviewer's rating.

**Reviewer Scores:**

- Reviewer V1cN: The reviewer is already positive. As their questions are reasonably answered in the rebuttal, I think they would maintain the score.
- Reviewer NyUx: The reviewer is already positive. As their questions are reasonably addressed, I think they would maintain the score.
- Reviewer ychT: I tend to think that they would increase only slightly, so that their rating would bring the paper below the bar.
- Reviewer Y5MW: Some key questions raised by the reviewer is answered in the rebuttal. However, lack of empirical evaluation stands. I therefore believe that the reviewer would not become positive and may only increase the score marginally.

---

### Decision · Program_Chairs · 2026-01-26

Reject